# Impact-based early warning of mass movements - A dynamic spatial modelling approach for the Alpine region

Stefan Steger<sup>1</sup>, Raphael Spiekermann<sup>1</sup>, Mateo Moreno<sup>2</sup>, Sebastian Lehner<sup>1,3</sup>, Katharina Enigl<sup>1,3</sup>, Alice Crespi<sup>4</sup>, Matthias Schlögl<sup>1,5</sup>

Correspondence to: Stefan Steger (stefan.steger@geosphere.at)

Abstract. Early warning systems play a crucial role in mitigating the impacts of severe weather events and related hazards. Traditional systems typically focus on meteorological forecasts and often do not account for the potential consequences that may follow, unlike impact-based approaches. In densely populated mountainous regions, such as the Alps, heavy precipitation frequently causes damaging mass movements. Since mass movement impacts ultimately result from a complex interplay of meteorological, geo-environmental, and socio-economic factors, warnings based solely on precipitation may have limited effectiveness. This study introduces a dynamic, spatially explicit modelling framework for impact-based early warning of precipitation-induced mass movement processes, tailored to three movement types: slides, flows, and falls. The framework integrates predisposing, preparatory, and triggering conditions, combining geo-environmental, meteorological, and exposure data to estimate daily impact potential across the Alpine region (450,000 km²). Using Generalized Additive Mixed Models (GAMMs), the approach captures non-linear relationships between impacts and predictors, ensuring interpretability and operational relevance. Beyond accounting for meteorological, geo-environmental, and exposure information, further key elements of the approach include incorporation of potential runout paths while maintaining a basin-based landscape representation, focusing model training on relevant terrain and time-periods to avoid trivial predictions, generating interpretable outputs, and demonstrating applicability through time-series predictive maps derived from hindcasting and "whatif" scenarios. Results highlight the strong operational potential of slide- and flow-type models, while the fall-type model exhibits limited usability for early warning, due to its low sensitivity to short-term weather conditions. Beyond early warning, the framework demonstrates broad applicability for analysing spatio-temporal patterns, conducting trend analyses, and assessing climate change impacts. This research advances the fields of landslide prediction and impact-based warning by providing a transferable and generalizable approach, offering actionable insights for disaster risk reduction and climate adaptation strategies.

<sup>&</sup>lt;sup>1</sup>GeoSphere Austria, Vienna, Austria

<sup>&</sup>lt;sup>2</sup>OpenGeoHub Foundation, Doorwerth, The Netherlands

<sup>&</sup>lt;sup>3</sup>Department for Meteorology and Geophysics, University of Vienna, Vienna, Austria

<sup>&</sup>lt;sup>4</sup>Center for Climate Change and Transformation, Eurac Research, Bozen/Bolzano, Italy

<sup>10 5</sup>Department of Landscape, Water and Infrastructure, BOKU University, Vienna, Austria

## 1 Introduction

Early warning systems play an important role in reducing the impacts of severe weather events and associated hazards. Traditionally, these systems have been focused on meteorological forecasts over large areas, predicting critical atmospheric conditions such as strong winds, hail or heavy precipitation (Alfieri et al., 2012; Sun et al., 2014; Weyrich et al., 2018; Sivle et al., 2022). While effective in identifying potentially hazardous meteorological conditions in advance, they do not address potential negative consequences that may follow. To improve risk management and motivate more appropriate response, it is essential to assess how critical weather conditions may impact lives, livelihoods and property (WMO, 2015; Casteel, 2016; Weyrich et al., 2020). Thus, several national meteorological and hydrological services incorporate impact-oriented or impact-based warnings to better inform decision-makers and the public about the potential adverse effects of meteorological events (Uccellini and Hoeve, 2019; Kaltenberger et al., 2020; Potter et al., 2021; Sivle et al., 2022).

Impact-oriented approaches primarily rely on meteorological data, emphasizing expected weather conditions while using

Impact-oriented approaches primarily rely on meteorological data, emphasizing expected weather conditions while using expert judgment to infer potential impacts. In contrast, impact-based approaches offer a more quantitative perspective on potential negative consequences by explicitly integrating meteorological information with data on geo-environmental conditions and socio-economic factors (e.g., exposure and vulnerability). The transition from meteorologically focused to impact-based warnings is encouraged by the World Meteorological Organization and international frameworks, such as the Sendai Framework for Disaster Risk Reduction and the Early Warnings for All initiative of the United Nations, which advocate for integrated risk-oriented strategies (WMO, 2015; Potter et al., 2021). Approaches for deriving impact-based warnings have been demonstrated for various phenomena, such as hailstorms, windstorms, droughts and flood-related assessments (Merz et al., 2020; Alfieri et al., 2024; Najafi et al., 2024; Schmid et al., 2024; Oh and Bartos, 2025). However, to our knowledge, no operational examples currently exist where impact potential is spatially modelled at high temporal resolution for hillslope processes, such as mass movements across large areas.

Even the step preceding impact-based forecasting, namely predicting the geomorphic phenomenon itself, remains a major challenge. Although some operational landslide warning systems exist, they are still widely considered complex and uncertain (Alfieri et al., 2012; Piciullo et al., 2018; Guzzetti et al., 2020). The complexity stems from the underlying intricate interplay between meteorological conditions and geo-environmental drivers, which is difficult to capture using available data over large areas. Their occurrence involves processes operating across different timescales, from static predisposing conditions to dynamic preparatory factors and short-term triggers (Crozier, 1989). For instance, slide-type movements are influenced by static factors, including slope morphology and subsurface material type, while preparatory factors, like antecedent rainfall or seasonal vegetation changes, can affect how a hillslope reacts to short-term disturbances, such as intense precipitation (Luna and Korup, 2022; Steger et al., 2023). Other mass movement types, like flow-type or fall-type movements, may be driven by distinct causal factors and therefore require tailored modelling approaches (Loche et al., 2022). Differentiating between movement types and their respective drivers is often overlooked in large-area assessments or, when acknowledged, remains difficult to address due to resource and data limitations (Günther et al., 2014; Caleca et al., 2025).

A further challenge lies in assessing landslide impacts, as triggered landslides travel downslope and can damage assets located far from their release zones. Consequently, spatially explicit landslide runout models are essential for impact and risk evaluations (Horton et al., 2013; Mergili et al., 2015; Wichmann, 2017). However, many spatially explicit large-area landslide assessments focus primarily on release zones or do not distinguish between release areas and downslope movement paths. This limitation reduces the ability of conventional landslide susceptibility models to evaluate impacts within the potential reach of landslides, such as flat terrain at the base of hillslopes (Mergili et al., 2019; Lima et al., 2023; Marchesini et al., 2024). Beyond the widespread use of static landslide susceptibility models that indicate where landslides may occur (Reichenbach et al., 2018) and empirical rainfall thresholds that focus on critical precipitation conditions for their initiation (Brunetti et al., 2010; Segoni et al., 2018; Peres and Cancelliere, 2021; Kaitna et al., 2025), recent efforts have increasingly focused on datadriven space-time landslide modelling (Lombardo et al., 2020). These models dynamically estimate critical landslide conditions across space and time by integrating geo-environmental factors with dynamic conditions, most notably weatherrelated variables (Stanley et al., 2021; Maraun et al., 2022; Ahmed et al., 2023; Moreno et al., 2024; Nocentini et al., 2024; Mondini et al., 2025). Most of these approaches focus on critical conditions within landslide release zones, without explicitly accounting for downslope landslide propagation or the assets located in potentially affected areas. Thus, despite being dynamic and spatially explicit, they lack key components essential for impact-based warning, such as information on elements at risk, which are explicitly addressed in landslide risk or exposure assessments (Dai et al., 2002; Corominas et al., 2013; Lin et al., 2023; Marchesini et al., 2024; Caleca et al., 2025). So far, landslide risk or exposure assessments for large areas only occasionally incorporate dynamic factors, such as land cover changes or climate trends (Farvacque et al., 2019; Ozturk et al., 2022; Lin et al., 2023). However, their utility for early warning is limited, as they fail to capture short-term meteorological

From a technical viewpoint, flexible algorithms and software tools offer extensive capabilities for processing large and heterogeneous datasets to create space-time models over large areas. However, their effectiveness in assessing and predicting environmental hazards and risks often remains limited by the quality and completeness of the input data used to train these models (Ardizzone et al., 2002; Guzzetti et al., 2006; Reichenbach et al., 2018). In fact, data limitations are frequently identified as a key barrier to more detailed large-area landslide modelling (Günther et al., 2014; Broeckx et al., 2018; Caleca et al., 2025). For instance, data-driven landslide models often have limited explanatory power due to spatially incomplete and temporally inconsistent landslide training data, which can lead to biased results. Targeted sampling and modelling strategies are required to adequately address these challenges or, at the very least, careful model interpretation (Steger et al., 2021; Caleca et al., 2025; Luna et al., 2025). Avoiding a high raster resolution or using alternative landscape representations, such as slope units or basins, can help reduce the impact of inaccuracies in landslide inventories on modelling results (Alvioli et al., 2016; Woodard et al., 2024). However, such generalizations can obscure important details. For instance, in landslide risk or exposure assessments, infrastructure or populations located within a mapping unit but well beyond the actual reach of potential landslides may be erroneously counted as exposed, simply because the entire unit is considered susceptible (Caleca et al., 2025).

variability and therefore cannot provide predictions at daily or sub-daily scales.

100

105

110

The sampling strategy used to represent typical landslide and non-landslide conditions strongly influences subsequent modelling results (Guo et al., 2024). Including data from irrelevant trivial terrain or trivial time periods, such as flat areas or dry conditions, can lead to landslide models that learn and reproduce overly simplistic patterns while inflating model performance scores. For example, such apparently well-performing models may mainly separate steep from flat areas or rainy from dry days, missing conditions critical for decision-making (Steger and Glade, 2017; Steger et al., 2023).

Model interpretability becomes particularly important when outputs are used to support decision-making (Schlögl et al., 2025), especially in early warning contexts (Vinuesa and Sirmacek, 2021; Reichstein et al., 2025). Transparent and interpretable outputs facilitate effective communication with stakeholders and foster trust in model results (Schlögl et al., 2024). For modellers, interpretable outputs are equally valuable, as they support plausibility checks and iterative refinements throughout the development process (Lombardo et al., 2020; Collini et al., 2022; Nocentini et al., 2023; Caleca et al., 2024).

In summary, many large-area landslide assessments remain of limited use for impact-based warning purposes. This is often due to their static nature, which fails to capture short-term meteorological variability, their exclusive focus on initiation zones while overlooking downslope runout areas and exposed assets, and limitations related to data availability and modelling design. This study addresses these gaps by developing and evaluating interpretable data-driven models to dynamically assess the impact potential of precipitation-induced mass movement processes across the Alpine region at daily scale. The models are designed to assess the impact potential of three movement types, namely slides, flows and falls, on infrastructure.

#### 115 Key features of the proposed approach include:

- Integration of impact drivers: Capturing the interplay of meteorological, geo-environmental and exposure data by integrating predisposing, preparatory and triggering conditions.
- Incorporation of runout paths: Accounting for movement paths while maintaining a generalized landscape representation by incorporating pixel-based mass movement runout information into basin-based model training.
- Tailored sampling strategies: Avoiding oversimplified patterns by constraining model training to non-trivial terrain and time-periods.
  - Model interpretability: Generating explainable outputs that support process-oriented plausibility checks and effective communication of results.
- Demonstrating suitability for impact-based warning systems: Producing maps and animations for hindcasting and scenario based examples.

The modelling framework presented in this study is designed as a generalizable and transferable approach for dynamically estimating the impact potential of fast-onset hazards within an impact-based warning context.

# 2 Study area

This study focuses on the Alpine Space as delineated by the Interreg Alpine Space Programme of the European Union (Fig. 1), an area that encompasses approximately 450,000 km² across seven countries: Austria, France, Germany, Italy, Liechtenstein, Slovenia and Switzerland.

Figure 1: Location and topography of the study area, the Alpine Space, as delineated by the Interreg Alpine Space Programme.

- The study area represents a distinctive and heterogeneous mountainous system within Europe, encompassing high alpine terrain, peri-alpine lowlands and densely populated valleys. Elevations range from sea level in the southern coastal areas to elevations of more than 4,800 m above sea level in the Western European Alps (Fig. 1). The considerable elevation gradient along the Alpine arc results in high relief energy, which plays a key role in conditioning slope instability.
- The hillslope basins in the Alpine Space are lithologically diverse (Donnini et al., 2020). Sandstone and claystone formations represent the dominant class within approximately 34% of the basins used in this study, followed by mixed carbonate rocks making up 28% and pure carbonate rocks approximately 23%. Igneous rocks, both acid and mafic types, cover 10%, while metamorphic rocks account for approximately 5%. This lithological diversity reflects the complex geological context of the

region and plays a significant role in shaping soil development, hydrological processes and slope stability (Donnini et al., 2020; Sidle and Ochiai, 2006).

Land cover also plays an important role in mass movement occurrence through various hydrological and mechanical effects (Crozier, 1989). Land cover data shows that the study area includes a considerable proportion of forested land, with broadleaf and coniferous tree cover accounting for approximately 38% of the area (Malinowski et al., 2020). Agricultural lands, including cultivated areas and vineyards, account for approximately 23%, while grasslands and shrub-dominated areas, including herbaceous vegetation, sclerophyllous cover and moorlands, comprise around 27%. Artificial surfaces, including urban infrastructure, represent a relatively small portion at 4%. Wetlands, bare natural surfaces, permanent snow-covered areas and water bodies together contribute around 8% (Malinowski et al., 2020).

A range of climate-related factors contribute to mass movement occurrence, with precipitation and temperature being the most influential, while snowmelt, seasonal variability and specific weather types also play important roles (Gariano and Guzzetti, 2016). Climatically, the region exhibits a high degree of variability in both temperature and precipitation. Temperature variations are pronounced due to the complex topography between valleys, ridges and mountain tops, as well as diverse climatic influences. These include maritime influences from the Atlantic Ocean and Mediterranean Sea, and continental effects from Eastern Europe. Precipitation is shaped mainly by the prevailing meandering westerly wind belt, moisture sources from the Atlantic and Mediterranean, and characteristic pathways of low-pressure systems. In addition, seasonal variations and local orographic effects also contribute to precipitation dynamics (Schär et al., 1998; Hofstätter and Chimani, 2012; Ménégoz et al., 2020).

Across the Alps, various elements at risk are exposed, such as population, infrastructure and critical assets (Alfieri et al., 2012; Günther et al., 2014; Keiler and Fuchs, 2016; Caleca et al., 2025). From a socio-economic perspective, the Alpine Space is home to over 80 million people and includes both major European metropolitan centres and remote rural regions. The area contains densely populated transport corridors, particularly within narrow valley systems where natural hazard-prone terrain overlaps with human settlements, leading to elevated exposure levels. Thus, development in the Alpine region is intrinsically coupled with natural hazard risk, with landslides representing an important component (Fuchs et al., 2017; Schlögl et al., 2019).

#### 3 Data





Topographic information was sourced from a 20 m Digital Terrain Model (DTM) covering the entirety of Europe (Sonny, 2023; Fig. 1). The DTM served as the foundation for delineating basins, for mapping potential landslide process paths and for deriving morphometric predictor variables. Lithological data was derived from the Alpine Geo-Lithological Map (Alpine-Geo-LiM) developed by Donnini et al. (2020). This map offers a simplified and harmonized lithological classification of the Alpine arc at a scale of 1:1,000,000 and was derived from national geological maps of Austria, France, Germany, Italy, Slovenia and Switzerland. Information on land cover was obtained from the Pan-European land cover and land use map published by Malinowski et al. (2020). The dataset provides consistent land cover information for the year 2017 across Europe and is based

on an automated classification of multi-temporal Sentinel-2 imagery. Meteorological data for the period 2005 to 2020 were obtained from the Copernicus European Regional ReAnalysis (CERRA), which provides high-resolution (5.5 km) gridded climate data for Europe. Temperature variables were derived from CERRA, while precipitation variables were taken from its Land component (CERRA-Land) (Ridal et al., 2024). Temperature was available at 3 hourly temporal resolution from CERRA analysed fields. These were aggregated to daily min/mean/max fields and then reprojected to EPSG:32632. Precipitation data from CERRA-Land are available directly as daily analysed fields, hence no further preprocessing except for the reprojection was needed. This dataset has already demonstrated promising potential for linking meteorological conditions to landslide occurrence in the Alps (Crespi et al., 2025). Exposure data was sourced from building and infrastructure layers available in OpenStreetMap (OSM), a widely used source of volunteered geographic information that provides comprehensive and regularly updated cartographic data, with the highest data completeness observed in Europe (Goodchild, 2007; Zhou et al., 2022).

The datasets cover the entire Alpine Space, while mass movement event data for model training was available for Austria and South Tyrol, a region representative of the environmental variability across the Alpine Space. This training area comprises 3,696 basins (Fig. 3; cf. Section 4.2.1) and, in addition to time-stamped event data, offers substantial diversity in geology, topography, land cover and precipitation, providing a robust basis for developing a model with spatial transferability across the Alpine Space.

Mass movement data were compiled from national inventories and filtered to include only precipitation-induced events (2005–2020) with known dates and documented infrastructure damage, ensuring relevance for impact-based analysis. Sources include the national GEORIOS inventory from GeoSphere Austria (Themessl et al., 2022), which provides time-stamped records of slides, flows and falls across Austria. From the event inventory of the Austrian Torrent and Avalanche Control (WLK), only flow-type events were extracted (BMNT, 2018; Heiser et al., 2019). The data for Northern Italy, encompassing slides, flows and falls, originate from the provincial implementation of the national landslide inventory IFFI (Inventario dei Fenomeni Franosi in Italia) and were accessed via the IdroGeo platform (Iadanza et al., 2021). While these inventories are subject to known limitations, they offer a valuable and comprehensive source of time-stamped mass movement occurrences enriched with contextual attributes. A notable characteristic, particularly limiting for purely process-oriented studies, is their tendency to underrepresent slope instabilities that did not cause damage (Trigila et al., 2010; Heiser et al., 2019; Steger et al., 2021). However, this focus on damage-causing events aligns well with the objectives of this study, which specifically targets an impact-based perspective.

# 4 Methods




#### 4.1 Methodical framework

This research developed three daily-scale impact-based models, each specifically tailored to slide-, flow- and fall-type mass movements. Figure 2 presents the methodological framework of this study, with further details provided in Sections 4.2 to 4.4.

Figure 2: Methodical framework of this study.

In summary, the study area was divided into 17,872 basins. Within each basin, areas potentially affected by a mass movement, hereafter referred to as potential process areas (PPAs), were delineated for each movement type to identify terrain where impacts are possible (Section 4.2.1). Movement type-specific binary response variables, representing presence and absence of impacts, were then generated using a spatio-temporal sampling scheme that restricts observations to relevant basins and relevant periods (Section 4.2.2). Predictor variables representing geo-environmental and exposure characteristics were aggregated within the PPAs of each basin, while daily meteorological data was aggregated to the basins using areal weighting (Section 4.2.3). Mass movement impacts were then modelled separately for each movement type as a function of meteorological, geo-environmental and exposure factors (Section 4.3). The data-driven models were evaluated using quantitative metrics and plausibility checks and then applied across the entire Alpine domain to generate daily impact predictions, visualized as maps and animations (Section 4.4).

## 4.2 Data preparation



# 220 4.2.1 Basin Delineation and Mapping of Potential Process Area

The study area was first subdivided into 17,872 hydrological half-basins using the *r.watershed* module in GRASS GIS 8.4 based on a resampled 50 m DTM (Neteler et al., 2012). Throughout the paper, these spatial units are referred to as basins. The



basins have a median size of 2,020 ha, with a lower threshold of 200 ha (Fig. 3). Polygons smaller than 200 ha were successively merged with adjacent units, first prioritizing the longest shared boundary, and subsequently the largest neighbouring area. Then, PPAs were delineated separately for each movement type to account for areas where mass movement impacts can reasonably be expected (Fig. 3a and 3b). This step was essential to focus the analysis within each basin on non-trivial terrain, thereby excluding irrelevant features beyond the potential reach of mass movements. Given the Alpine-wide scale, the Gravitational Process Path (GPP) model (Wichmann, 2017) was used to identify potential runout paths based on the available 20 m DTM. Our approach combined the angle of reach concept with stochastic random walk routing along the surface of a DTM, balancing realism and computational efficiency to approximate areas potentially affected by mass movements. Although less precise than site-specific runout models, the derived PPAs were considered to serve as meaningful spatial constraints for further analyses.

Figure 3: Basins (red) and potential process areas (PPA) for fall-type movements (brown) across the Alpine Space (a). Close-up of Bozen/Bolzano in northern Italy, showing process paths for fall-types that were used to delineate the PPA (b). Trivial terrain for fall-types (grey), the PPA for fall-types (white) and OSM building data (c). Note that geo-environmental and exposure variables





were aggregated to the basins based on their distribution within the PPAs (white areas in c). Overview of all PPAs within the example basin (d).

Parameterization for each movement type was derived from literature and the GPP model documentation. The angle of reach is an empirical parameter that primarily governs the resulting runout length, while runout behaviour was controlled by three parameters: slope weighting, divergence exponent and the persistence factor (Wichmann, 2017). For slide-type movements, an angle of reach of 22° was applied (D'Amboise et al., 2021), with a slope weighting of 30, a divergence exponent of 2.12 and a persistence value of 1.75 (Wichmann, 2017). Flow-type movements used an angle of reach of 11° (Horton et al., 2013), with the same slope weighting, divergence exponent and persistence values as slide-types. For fall-type movements, an angle 245 of reach of 29° was used (Menk et al., 2023), combined with a slope weighting of 60, a divergence exponent of 1.75 and a persistence of 1.3 (Wichmann, 2017).

The GPP model also requires spatially explicit information on potential release areas to serve as starting points for process path simulations. The potential release areas were also defined based on literature using conservative slope thresholds selected to capture a broad extent of likely initiation-prone terrain. From each potential release pixel, five random walks were initiated to capture some variability in downslope routing. For slide-types, slopes between 5° and 50° were considered potential release locations (Steger et al., 2021). Flow-type release areas were delineated on terrain with slopes greater than 15° (Horton et al., 2013), while fall-type release areas were mapped on slopes exceeding 35° (Dupire et al., 2020).

#### 4.2.2 Response variables: Representing occurrence and non-occurrence of mass movement impact

For each movement type, a binary response variable representing the presence and absence of mass movement impact was created using a structured spatio-temporal sampling scheme. All observations, presences and absences, were limited to "nontrivial" situations. Spatially, only basins where infrastructure overlapped with the PPA were considered and temporally, only days with at least moderate precipitation were included. Thus, subsequent modelling was constrained to observations representing conditions potentially relevant for impact-based warning.

In detail, presence data were prepared using three data sources, GEORIOS, WLK and IFFI (Section 3). Initially, all records of deep-seated movements or those lacking documented infrastructure impact were excluded, based on available inventory attributes. The remaining presence data were grouped into slide-, flow- and fall-type movements and spatially joined to their corresponding basins. Further filtering was applied to retain only events with a known day of occurrence between 2005 and 2020 that were associated with at least moderate precipitation conditions. Building on our previous work (i.e., Steger et al., 2023), a rainfall filter was applied using precipitation data from the event day and the preceding day. The inclusion of the preceding day was important to account for cases where e.g., rainfall occurred in the late evening, resulting in a mass movement that was initiated/recorded on the following day. An arbitrary threshold of 10 mm for daily precipitation was used to define a day with moderate precipitation amounts, applied to either day. Note that the number of presence basins is lower than the total number of presence observations used for modelling, since multiple days with registered mass movement occurrences may have been recorded within a basin during the 16-year period.

- Absence basins were randomly selected using a spatial distance constraint to minimize the selection of neighbouring basins and reduce spatial autocorrelation. Specifically, if basins had centroids within 10 km of each other, only one basin was selected. This was done while maintaining a one-to-one ratio between presence and absence basins for each movement type, which resulted in 2,028 basins for the slide-type (1,014 presences and 1,014 absences), 1,382 basins for the flow-type (691 each) and 1,098 basin for the fall-type (549 each).
- Absence dates were sampled from both absence and presence basins. For each basin, 50 random dates between 2005 and 2020 were initially drawn, creating a large pool of absence days. To reduce potential temporal autocorrelation, absence dates within the same basin were excluded if they fell within 30 days of any registered mass movement event. Additionally, if multiple absence dates within the same basin were less than 30 days apart, only one date was retained. Based on this, further downsampling was applied to distribute absence dates evenly across months and years, ensuring equal representation of all time periods, in analogy to Steger et al. (2023). From this pool of evenly distributed and temporally spaced absence dates, only days meeting the same rainfall threshold as the presence data were retained to create a sample of at least moderately rainy days not associated with recorded mass movement occurrences. In summary, the final datasets comprised 7,900 observations for the slide-type models (2,077 presences and 5,823 absences), 5,463 observations for flow-type models (1,386 presences and 4,077 absences) and 4,069 observations for fall-type models (647 presences and 3,422 absences).

# 4.2.3 Predictor variables: Characterizing potential drivers of mass movement impact

A range of predictor variables was compiled to represent the geo-environmental, exposure-related and meteorological/climatic conditions used to estimate mass movement impact. The following table shows the variables used to describe these conditions, along with their classification, spatial aggregation methods and spatio-temporal characteristics.

Table 1: Overview of predictor variables used for mass movement impact modelling, including their classification, spatial aggregation methods and spatio-temporal characteristics. Note that the variables in bold were included as candidates within the initial model setup (cf. Section 4.3).

| Category          | Variable (v) / Variable class<br>(c) | Spatial aggregation     | Туре             |
|-------------------|--------------------------------------|-------------------------|------------------|
| Geo-environmental | Slope angle (v)                      | Mean in PPA             | Spatial & static |
| (Morphometry)     | Convergence (v)                      | Mean in PPA             |                  |
|                   | Aspect (v)                           | Mean in PPA             |                  |
| Geo-environmental | Land cover (v)                       | Dominant class in PPA   | Spatial & static |
| (Land cover)      | Deciduous forest (c)                 | Portion of class in PPA |                  |
|                   | Coniferous forest (c)                | Portion of class in PPA |                  |
|                   | Crop- & grassland (c)                | Portion of class in PPA |                  |
|                   | Bare surface & other (c)             | Portion of class in PPA |                  |
| Geo-environmental | Lithology (v)                        | Dominant class in PPA   | Spatial & static |
| (Lithology)       | Igneous (c)                          | Portion of class in PPA |                  |
|                   | Metamorphic (c)                      | Portion of class in PPA |                  |




|                 | Mixed carbonate (c)                | Portion of class in PPA    |                      |
|-----------------|------------------------------------|----------------------------|----------------------|
|                 | Pure carbonate (c)                 | Portion of class in PPA    |                      |
|                 | Sandstone-claystone (c)            | Portion of class in PPA    |                      |
| Exposure        | Number of buildings (v)            | Number of buildings in PPA | Spatial & static     |
|                 | Transport infrastructure (v)       | Portion of "road" cells in |                      |
|                 |                                    | PPA                        |                      |
| Meteorological  | Short-term precipitation (v)       | Area-weighted spatial      | Spatial & daily      |
| (Precipitation) | Antecedent precipitation: 7-       | averaging                  |                      |
|                 | day, 14-day, 21-day, <b>30-day</b> |                            |                      |
|                 | precipitation (v)                  |                            | Spatial & static     |
|                 | Mean annual precipitation          |                            |                      |
|                 | 2005-2020 (v)                      |                            |                      |
| Meteorological  | Mean daily temperature (v)         | Area-weighted spatial      | Spatial & daily      |
| (Temperature)   | Binary: Day crossing 0 °C (v)      | averaging                  |                      |
|                 | Binary: Day below 0 °C (v)         |                            |                      |
| Others          | Day-of-Year (DOY)                  | -                          | Non-spatial & daily  |
|                 | Sampling Year                      | -                          | Non-spatial & yearly |
|                 | Sampling location (Basin-ID)       | -                          | Spatial & static     |

The static geo-environmental variables related to basin morphology were derived using *SAGA GIS* (Conrad et al., 2015) and the 20 m DTM. These variables were aggregated from the pixel level to the basin scale based on the respective PPAs. As a result, parameters, such as mean slope angle or the number of potentially exposed buildings, vary within the same basin, depending on the underlying movement type.

Land cover and lithology variables were based on reclassified versions of their original datasets (Donnini et al., 2020; Malinowski et al., 2020). The reduction in the number of classes aimed to increase the likelihood of obtaining interpretable and parsimonious models. Care was taken to ensure that each class was well represented within the training area, while also ensuring that no class occurring outside the training area lacked adequate representation in the training data. The four major land cover classes and five lithological classes are well represented within the training area and broadly distributed across the entire Alpine Space (Table 1).

Variables representing exposure were obtained from OSM and include buildings and transportation infrastructure. For buildings, individual point data were aggregated for each basin by summing the number of buildings within each PPA (Fig. 3c), resulting in approximately 19.8 million buildings within slide-type PPAs, 12.7 million in flow-type PPAs and 1.2 million in fall-type PPAs. Primary and secondary road networks and railways (e.g., highways, railways and paved roads) represented transport infrastructure, while trails and hiking paths were excluded. These data were aggregated for each basin by calculating the fraction of rasterised roads (20 m grid cells) within each PPA.

Temperature variables were derived from the 3 hourly analysed CERRA fields, with precipitation data sourced from CERRA-Land as daily fields. The 3 hourly temperature fields were aggregated to daily min/mean/max fields. These 5.5 km resolution data were aggregated to the basin level using spatial weights based on the degree of overlap between CERRA grid cells and







basin polygons. Following Steger et al. (2023), precipitation was represented using a short time window to capture "triggering" precipitation and longer time windows to account for antecedent precipitation. Triggering precipitation referred to the cumulative amount on the observation day and the preceding day, while antecedent precipitation was calculated over 7-, 14-, 21- and 30-day periods preceding the triggering precipitation. Mean annual precipitation over the 2005 to 2020 training period was included as a static baseline describing long-term spatial variability in precipitation. Daily mean temperatures and two binary indicators were used to represent potential temperature effects. The binary variables represent (1) whether the temperature crossed 0 °C on a given day, serving as a potential indicator for freeze-thawing and (2) whether the temperature remained continuously below 0 °C on a day, indicating potential frozen ground conditions. Finally, additional variables such as Day-Of-Year (DOY), sampling year and sampling location were included to account for seasonal patterns and group-level differences between sampling units.

#### 4.3 Modelling

The potential of mass movement impact was modelled as a function of geo-environmental, exposure-related and meteorological variables using Generalized Additive Mixed Models (GAMMs). Separate models were fitted for each movement type, using parameter sets specific to the associated PPAs (Fig. 3d). GAMMs extend Generalized Additive Models (GAMs) by incorporating random group-level effects, while retaining the strengths of GAMs in modelling complex non-linear relationships through smooth functions, accommodating diverse error distributions and maintaining high interpretability. These characteristics make GAMMs particularly well-suited to environmental modelling tasks (Zuur et al., 2009; Wood, 2017; Pedersen et al., 2019). Various versions of GAMs and GAMMs have been applied in data-driven landslide research (Goetz et al., 2015; Lombardo et al., 2018). In this study, the binary response was modelled using a binomial error distribution. The model was implemented using the *bam()* function from the *mgcv package* in R, which is specifically designed for modelling with large datasets (Wood et al., 2015; Wood, 2017). Variable selection was performed using an automated procedure based on double penalty shrinkage, which penalizes both the complexity of smooth terms to prevent overfitting and the null space to allow exclusion of non-informative predictors (Marra and Wood, 2011).

The modelling process began with a comprehensive initial variable setup (cf. variables in bold in Table 1) and proceeded iteratively, using automated variable selection throughout. To account for the hierarchical structure of the data, random intercepts were included for sampling year and sampling location. In the initial model, care was taken to avoid including multiple variables representing the same underlying effect. Specifically, only one variable for antecedent precipitation (based on a 30-day accumulation window) was included, while land cover and lithology were represented as categorical variables based on their dominant class within the PPAs. The resulting reduced models served as the basis for the next step, in which categorical land cover and lithology variables were supplemented with their continuously scaled counterparts (cf. Table 1) to assess whether additional information on the proportion of specific classes added value, both in terms of statistical significance (p-value 






automated selection procedure, contributing significantly and exhibiting a plausible positive relationship, as higher proportions of bare surface within the PPA can reasonably be associated with increased rockfall activity.

The final step involved testing whether the initially selected 30-day accumulation window for representing antecedent precipitation was the best choice. This was evaluated using cross-validation (cf. Section 4.4.1), comparing the predictive performance of the 30-day model against alternative accumulation periods (7, 14 and 21 days), as well as against a null model that excluded antecedent precipitation entirely. As a result, one final model was selected for each movement type. For predictive mapping across the Alpine Space and performance assessment, predictions were based on fixed effects only, with random effects averaged (Wood, 2017).

## 4.4 Model evaluation

The modelling results were evaluated using a plurality of approaches. Model performance was assessed through cross-validation (Section 4.4.1). Variable importance assessment and inspection of partial effects provided insights into model relationships and supported plausibility checks (Section 4.4.2). Finally, the models were applied across the entire Alpine domain to generate daily spatial predictions for each basin (Section 4.4.3).

# 4.4.1 Assessing model performance

The fitting performance of each model was evaluated on the training data by visualizing receiver operating characteristic (ROC) curves and calculating the area under the ROC curve (AUROC). The ROC curve illustrates the trade-off between the true positive rate (TPR) and the false positive rate (FPR) across different classification thresholds derived from predicted probabilities ranging from 0 to 1. The AUROC is a standard metric in binary classification that provides a single-number summary of the ROC curve, reflecting the ability of the model to distinguish between presence and absence observations. An AUROC of 1 indicates perfect discrimination, while a value of 0.5 corresponds to random guessing (Metz, 1978; Fawcett, 2006).

To evaluate predictive performance and identify the optimal antecedent rainfall time window, we employed 5-fold cross-validation with basin-based partitioning, repeated 10 times. Specifically, in each fold, 80% of randomly selected basins were used as the training set to fit the model, while the remaining 20% served as the test set for deriving the AUROC. Within each repetition, every basin was used once as a test basin and four times as part of the training set. Repeating this process across 10 independent random partitions resulted in 50 AUROC values per model (5 folds time 10 repetitions).

## 4.4.2 Assessing modelled relationships

To enable model interpretation and plausibility checks of the internal model behaviour, we used variable importance assessment alongside visualization of partial effects. Variable importance, which quantifies the relative contribution of each predictor, was assessed using a permutation-based approach implemented in the R package *vip* (Greenwell et al., 2020). Specifically, for each predictor, values were randomly shuffled across the dataset, breaking the relationship with the response


variable. The impact of this permutation on model performance was measured by the resulting decrease in AUROC. This procedure was repeated 100 times per predictor using Monte Carlo replications. Predictors causing a larger reduction in AUROC were interpreted as having higher relative importance in the model.

Partial effects plots complement variable importance assessment by providing insights into the modelled relationships at the single-predictor level. In the context of GAMs, partial effect plots graphically represent the estimated smooth functions that describe the relationship between the response and predictor variables (Simpson, 2024). In our study, these plots illustrate how the modelled effect of each predictor varies across its observed range, showing the contribution of that predictor to the overall prediction.

# 4.4.3 Creation of predictive maps

A series of predictive maps and animations were generated by applying the three models across the Alpine Space. Model predictions, expressed as probability scores, were transferred to the basin level on a daily scale, excluding random effects and assigning a value of zero to trivial basins where no impacts are expected (i.e., where the PPA is absent or does not overlap with relevant infrastructure). In these maps, the probability scores do not represent absolute occurrence probabilities, but conditional probabilities influenced by the class balance in the training data (i.e., presence to absence ratio). Higher scores 390 denote stronger resemblance to impact conditions, whereas lower scores indicate conditions typical of non-impact observations (cf. Section 6.4 for guidance on interpretation and relevance for operational implementation). As the model predictions are space- and time-specific, predictive maps were produced for selected constellations of dynamic variables. In this context, both "what-if" scenarios and hindcasting were employed to visualize model behaviour under hypothetical and real conditions, respectively.

In analogy to Moreno et al. (2024), a range of "what-if" scenarios were produced to depict how the model responds when subjected to controlled, hypothetical input settings. Specifically, a stepwise increase in precipitation, introduced through spatially uniform alterations of daily precipitation amounts, was used to evaluate how the estimated probability of mass movement impact varies across the Alpine region. Hindcasting involves simulating and analysing past conditions using a model originally designed to predict future outcomes. It is commonly employed to validate predictive models by comparing their outputs with known historical situations, thereby assessing their reliability and plausibility (Clement, 2011; Ozturk et al., 2021; Moreno et al., 2024). In our study, hindcasting was applied to well documented past events to explore model behaviour using historical input data and to demonstrate its potential for application in an impact-based early warning context.

# **5 Results**


# **5.1 Model performance**

The ROC curves (Fig. 4a) indicate relatively high fitting performance for all three models, particularly if considering that easy-405 to-classify observations were a-priori excluded. The flow-type model achieved the highest AUROC (0.90), followed by the

slide-type (0.87) and fall-type (0.82) models. Points along each curve indicate the closest point to the top-left of the ROC space, representing the optimal thresholds that balance misclassification rates and best separate the two classes (Schisterman et al., 2005). At these thresholds, the slide-type model reached a true positive rate (TPR) of 0.78 and a true negative rate (TNR) of 0.81, the flow-type model achieved a TPR of 0.85 and TNR of 0.78 and the fall-type model a TPR of 0.78 and TNR of 0.72.

Figure 4: Model fitting performance and optimal cut-points based on the closest point to the top-left corner of the ROC space (a). Predictive performances from basin-based cross-validation across different antecedent precipitation time windows are shown for the slide-type model (b), flow-type model (c) and fall-type model (d). Final models were constructed using the best-performing time windows: 30 days for slide-type, 21 days for flow-type and 14 days for fall-type.



Cross-validation (Figs. 4b–d) was used to evaluate predictive performance and the influence of different antecedent precipitation time windows (7, 14, 21 and 30 days), alongside a null model without this variable. Median AUROCs were only slightly lower than their corresponding fitting performance (Fig. 4a), indicating a high predictive skill and limited model overfitting. For the slide-type and flow-type models, AUROC values increased more markedly with the inclusion of antecedent precipitation, whereas the fall-type model exhibited minimal variation across time windows, suggesting a weaker sensitivity to antecedent precipitation conditions.

The optimal time windows selected for the final models were 30 days for the slide-type model (median AUROC = 0.86), 21 days for the flow-type model (median AUROC = 0.89) and 14 days for the fall-type model (median AUROC = 0.80). Variability in predictive performance, expressed as the interquartile range (IQR) of AUROCs for the final models, can serve as an indicator of model stability, with higher IQRs denoting less consistent performance and greater uncertainty in generalization (Petschko et al., 2014). This variability was lowest for the flow-type models (IQR = 0.02), followed by the slide-type models (IQR = 0.03) and highest for the fall-type models (IQR = 0.05). While generally low across all three models, the variability indicates greater uncertainty in fall-type predictions while underscoring the relatively higher stability of the flow- and slide-type models.

#### 430 **5.2 Modelled relationships**

Permutation-based variable importance assessment revealed distinct patterns in the key factors influencing mass movement impact across the three models. The slide-type and flow-type models emphasize a combination of meteorological variables and static characteristics related to terrain morphology and exposure, whereas the fall-type model was dominated by static spatial variables, especially those related to exposure (Fig. 5).



Figure 5: Permutation-based variable importance for the final three models, expressed as the decrease in AUROC after permuting each variable 100 times.

For the slide-type and flow-type models, short-term precipitation, which served as a proxy for triggering rainfall, was the most influential variable, followed by mean annual precipitation. For the slide-type model, the next most important variables were mean slope angles, antecedent precipitation, the number of buildings in the PPA and the dominant lithology class within it. The flow-type model ranked mean daily temperature as the third most important predictor, followed by slope angle, antecedent precipitation and the number of buildings in the respective PPA. In contrast, the fall-type model showed a markedly different pattern, with static spatial factors dominating the top five ranks and short-term precipitation emerging only as the sixth most important variable. The most important predictors were exposure related, particularly the number of buildings and the extent of transport infrastructure within the fall-type PPA. Additionally, lithology, slope angle and the portion of bare surface were identified as relatively important factors.





Partial effect plots provided detailed insights into the modelled relationships between individual predictors and estimated impact potential at the single-predictor level (Fig. 6, 7, 8). When interpreting these plots, it is important to consider the relative importance of each predictor within the model (Fig. 5), distinguishing variables that are highly influential from those with lesser impact. For the slide-type model, the most important variable, namely short-term precipitation, showed a positive, nonlinear association with estimated mass movement impact (Fig. 6a). The increase was steep and approximately linear up to 100 mm, after which the effect began to level off. The smooth term for antecedent precipitation indicated that lower precipitation amounts within the previous 30 days were associated with reduced estimated impact probabilities (Fig. 6b). Very high antecedent precipitation amounts exceeding 400 mm did not correspond to further increases in impact probability. Consistent with Steger et al. (2024), a clear negative relationship with mean annual precipitation was observed (Fig. 6c), indicating that basins with generally drier conditions tend to respond more strongly to potential triggering events than basins adapted to wetter conditions. The potential for slide-type impact was higher in basins characterised by steeper terrain (Fig. 6d) and high exposure (Fig. 6e, f), indicated by the number of buildings and density of transport infrastructure within the potential slide-type terrain (PPA). Seasonal variation, modelled using a circular DOY effect, indicated that periods with reduced vegetation effects (i.e., winter, early spring) were associated with higher impact probabilities (Fig. 6g), noting that temperature and precipitation effects are accounted for by other variables in the model. Temperature variables revealed that particularly cold conditions below 0 °C corresponded to a lower impact potential, whereas hot days with mean daily temperatures exceeding 20 °C were associated with increased impact probabilities, suggesting that warmer "rainy" days, when convective events are more likely due to increased potential instability (Giorgi et al., 2016), may increase impact likelihood (Fig. 6h, k). Land cover and lithology showed partially significant differences among their classes, with basins dominated by deciduous forest and mixed carbonate lithology estimated to be more prone to slide-type impacts (Fig. 6i, j).

Figure 6: Partial effects for the slide-type model showing estimated smooth functions for continuous predictors and factor effects for categorical predictors, illustrating how each variable relates to modelled slide-type impact potential. Shaded bands indicate 95% confidence intervals for smooth terms, while points with lines represent estimated effects and confidence intervals for factor levels. All effects are centered around zero. Higher values indicate a higher estimated likelihood of class membership (impact = yes), meaning that the higher the value, the higher the estimated probability of a slide-type impact.

The flow-type models showed several patterns similar to those of the slide-type models. There was a clear positive relation between short-term precipitation and landslide impact (Fig. 7a), increased impact potential with higher antecedent precipitation (Fig. 7b) and a negative association with mean annual precipitation (Fig. 7c). Steeper slope angles and a higher number of buildings within the potential flow-type terrain (PPA) were further associated with higher impact potential (Fig. 7d, f). Seasonal effects, possibly related to vegetation periods, were again apparent (Fig. 7g). Notably, analogous to the slide-type model, mean daily temperature, the second most important predictor in the flow-type model, showed a strong positive relationship, likely reflecting the influence of rainfall-type, with warmer convective days increasing flow-type impact likelihood (Fig. 7h).


Figure 7: Partial effects for the flow-type model showing estimated smooth functions for continuous predictors and factor effects for categorical predictors. Shaded bands indicate 95% confidence intervals for smooth terms, while points with lines represent estimated effects and confidence intervals for factor levels.

The fall-type model showed generally plausible relationships with meteorological variables (Fig. 8), although these variables were previously identified as less influential (Fig. 5). The two most important variables were static exposure factors, both exhibiting a positive relationship with estimated impact potential: when few buildings or a low density of transport infrastructure are present within the potential fall-type terrain (PPA), estimated impact probabilities are generally low (Fig. 8g, h). Slope angles within the fall-type PPAs were considerably steeper than those in the slide- or flow-type PPAs (cf. x-axis ranges in Figs. 8d vs. 6d and 7d). For fall-type models, impact potential increased continuously with slope steepness, whereas slide- and flow-type models showing a saturation effect at higher slope angles. Basins dominated by igneous and metamorphic rocks were estimated to be most prone to fall-type impacts (Fig. 8j) and a higher proportion of bare surface within PPAs further increased impact potential (Fig. 8f).

Figure 8: Partial effects for the fall-type model showing estimated smooth functions for continuous predictors and factor effects for categorical predictors. Shaded bands indicate 95% confidence intervals for smooth terms, while points with lines represent estimated effects and confidence intervals for factor levels.

# 5.3 Maps and visualizations




To explore model behaviour under controlled hypothetical conditions, "what-if" scenarios were visualized. The maps in Fig. 9 show model responses to spatially uniform increases in short-term precipitation of 20 mm, 40 mm, 80 mm and 160 mm, while all other dynamic variables were held constant. The zoom-ins provide a more detailed view of the predictions at basin level. In line with the effects described in Section 5.2, slide-type and flow-type models responded strongly to increasing short-term precipitation (Fig. 9 top and middle rows). The fall-type model was comparatively less sensitive to this effect, with spatial patterns remaining relatively similar across precipitation scenarios (Fig. 9 bottom row). Because each basin was assigned the same short-term precipitation amount, the resulting prediction patterns reflect spatial differences in potential mass movement impact driven by geo-environmental and exposure characteristics. Across all models, the highest potential is concentrated along the Alpine arc, particularly where steep terrain coincides with higher densities of exposed assets. Slide-type models generally indicate the largest spatial extent of potentially impacted areas, covering not only the higher-relief Alpine core, but also extensive parts of the Alpine foreland, where relief energy is moderate (Fig. 9d). Flow-types affect smaller areas overall and were more confined to steep valleys within the Alpine arc (Fig. 9i). Potential impact areas due to fall-type movements were geographically the most restricted, with elevated values concentrated in the high-relief zones of the Western and Central Alps, with additional localised hotspots visible in the Eastern Alps (e.g., Dolomites) (Fig. 9n).




Figure 9: Visualization of "what-if" scenarios for slide-type model (a–e), flow-type model (f–j) and fall-type model (k–o) under spatially uniform increases in short-term precipitation. Shown are model predictions for "triggering" precipitation amounts of 20 mm, 40 mm, 80 mm and 160 mm, with all other dynamic variables held constant (antecedent precipitation = 50 mm, mean daily temperature = 20 °C, Day-of-year = 200, binary temperature = "Above 0 °C" and "Not crossing 0 °C"). The zoom-in panels depict the 80 mm "triggering" precipitation scenario. Animations showing precipitation increases in 10 mm increments from 10 to 200 mm are provided in the supplementary material for each movement type (S1-S3 at: https://doi.org/10.6084/m9.figshare.30271795.v1).

Hindcasting was used to analyse model predictions under known conditions and to demonstrate their potential for impact-based early warning. Storm Vaia (October 27 to 29, 2018) served as an example of a severe, large weather event that affected multiple European countries. During the storm, heavy rain and strong winds caused extensive forest damage, flooding, and numerous damage-causing mass movements across the Alpine region (Cavaleri et al., 2019; Giovannini et al., 2021; Antonetti et al., 2022).

Figure 10 shows the model predictions for October 29 and for the storm aftermath on November 2, along with precipitation variables. On October 29, the potential for slide-type (Fig. 10a) and flow-type impacts (Fig. 10b) was particularly high in northern Italy (Trentino-Alto Adige, Veneto, Liguria) and, to a lesser extent, in western Austria (Eastern Tyrol), southern Switzerland, southeastern France and northwestern Slovenia (Cavaleri et al., 2019; Menegatto et al., 2024). The precipitation maps indicate that this pattern was primarily driven by widespread, high precipitation amounts, with short-term precipitation being the dominant driver (Fig. 10d). Across most of the study area, the 30-day antecedent precipitation did not show exceptionally high values on October 29, except in the southwestern part (e.g., Liguria), where the precipitation front initially arrived (Fig. 10e). Notably, during this event, the highest precipitation occurred over mountainous terrain prone to mass

movement impact, while in low-susceptibility areas, such as flat areas in Germany, localized heavy rainfall (Fig. 10d) did not raise impact potential (Fig. 10a, b, c), illustrating that the model accounts not only for rainfall amounts alone.

By November 2, the estimated mass movement impact potential had generally decreased across the Alpine region, with more localised precipitation (Fig. 10i) still driving elevated predicted values in the southwest (Fig. 10f, g). Many basins in the center of the study area no longer received high precipitation. However, they still exhibited moderate impact potential, primarily due to very high antecedent precipitation (Fig. 10j), used as a proxy for soil moisture conditions. Comparing both days for the fall-type model (Fig. 10c and 10h) continued to demonstrate its relatively low sensitivity to weather dynamics.

Figure 10: Hindcast example for storm Vaia. Model predictions for October 29 (top row) and the aftermath on November 2 (bottom row) for three movement types: slide-type (a, f), flow-type (b, g) and fall-type (c, h). Corresponding precipitation variables for the same dates are shown, including short-term precipitation as used within all models (d, i) and antecedent 30-day precipitation as used for the slide-type model (e, j). Hindcast animations of this event are provided in the supplementary material (S4-S6 at: https://doi.org/10.6084/m9.figshare.30271795.v1).

In comparison to the storm Vaia, a more localised yet severe event struck the southeastern Alpine forelands from June 23 to 26, 2009. Triggered by thunderstorms associated with a low-pressure system over the Adriatic, the event led to more than 3,000 mass movements of the slide- and flow-type, severely affecting transportation infrastructure and prompting the evacuation of several houses. The Austrian district *Südosteiermark*, at the centre of the affected area, sustained extensive damage, leading to the declaration of a state of emergency. Disaster response and reconstruction costs for the state of Styria exceeded €13.4 million (Hornich and Adelwöhrer, 2010; Maraun et al., 2022). Figure 11 shows the hindcast of this event for the slide-type model, along with the districts associated with the highest number of private damage reports.




Figure 11: Hindcast example of a severe event in Styria, Austria. Model predictions from the slide-type model for June 25, 2009, are shown for the focus area of the event (a) and the entire Alpine Space (b), together with the variables representing short-term precipitation (c) and 30-day antecedent precipitation (d). Black polygons in (a) delineate the districts of Styria, with labels showing the districts linked to the most private damage reports: Südoststeiermark (SO, 600 reports), Leibnitz (LB, 324), Weiz (WZ, 287), Graz-Umgebung (GU, 205), Deutschlandsberg (DL, 95) and Fürstenfeld (FF, 62) according to Hornich and Adelwöhrer (2010). Hindcast animations of this event provided in the supplementary material are (S7 at: https://doi.org/10.6084/m9.figshare.30271795.v1).

According to Maraun et al. (2022), the precipitation event itself was severe, but not extreme. The exceptional mass movement occurrence has been attributed to the compounding effect of heavy rainfall combined with pre-moistening during the preceding winter and spring (Hornich and Adelwöhrer, 2010; Maraun et al., 2022; Mishra et al., 2023). Previous event-based landslide modelling for this area revealed that a five-day rainfall variable was the strongest predictor, suggesting that also shorter-term antecedent conditions contributed to the observed mass movements (Knevels et al., 2020). Figure 11 shows the slide-type model hindcast for June 25, with high estimated impact potential in the affected area largely matching the reported damage (Fig. 11a). For this model prediction, both high short-term precipitation (Fig. 11c) and elevated accumulated precipitation over the preceding 30 days (Fig. 11d) were key factors. Unlike the region-specific analyses by Maraun et al. (2022) the model did not capture pre-moistening at time scales beyond 30 days. Nonetheless, elevated wet conditions in June 2009 along the northern Alpine front led to widespread soil saturation and extensive flooding (Godina and Müller, 2009), likely influencing this mass movement event as well.

#### 6 Discussion

The discussion subsections 6.1 to 6.4 follow the main goals outlined in the introduction, while Section 6.5 addresses the general transferability of the approach and its potential application beyond impact-based early warning.







# 6.1 Capturing the interplay of impact drivers

The models were designed to estimate mass movement impact potential by integrating predisposing, preparatory and triggering conditions represented through meteorological, geo-environmental and exposure data. Separate models were generated for each movement type, because different landslide types are driven by distinct sets of conditions (Loche et al., 2022).

Regarding dynamic variables, the slide- and flow-type models are highly sensitive to daily changes, whereas fall-type models are heavily governed by static spatial variables. For slide- and flow-type models, the most important predictor of potential impact to infrastructure is short-term "triggering" precipitation. Antecedent precipitation, DOY, and temperature also have an important contribution, reflecting soil moisture, seasonality, and temperature effects. For instance, in the flow-type model, mean daily temperature ranked as the second most important predictor, with hot days showing the highest impact potential, likely due to an increased potential of convective activity (Giorgi et al., 2016), while days below 0°C were associated with the lowest impact likelihoods, possibly reflecting frozen ground effects on slope stability. The range of dynamic variables shows that the models captured effects across multiple temporal scales. This finding is in line with previous studies that show mass movement occurrence depends not only on conditions on the day of the event, but also on antecedent conditions over longer periods (Crozier, 1999; Mirus et al., 2018; Rosi et al., 2021; Maraun et al., 2022; Smith et al., 2023).

In this context, daily precipitation and temperature variables represented immediate conditions, antecedent precipitation captured cumulative effects over preceding weeks, and the DOY variable reflected seasonal influences, primarily related to vegetation (Steger et al., 2023), since other season-related meteorological effects were already accounted for. Although model performance was generally high (Fig. 4), better capturing environmental dynamics could further improve their explanatory power. For example, instead of relying on proxies, soil moisture data derived from simulations or hydrological modelling could better capture actual soil wetness conditions (Maraun et al., 2022), while incorporating information on sediment availability may further enhance the model (Heiser et al., 2023). Precipitation data at sub-daily resolution would likely capture actual rainfall triggering conditions more accurately, particularly for rapid flow-like events (Marra et al., 2016). In such case, sub-daily mass movement occurrence data should ideally be available to match this high-resolution data. When such detailed observations are lacking, aggregating sub-daily rainfall to daily scales by using e.g., the maximum three-hour rainfall intensity on a day, has been shown to be of value (Knevels et al., 2020; Smith et al., 2023).

Depending on the modelling context, potential enhancements in the dynamic component include snowmelt, vegetation seasonality, season-dependent precipitation windows, functional precipitation representations, and interactions with other geo-environmental variables (Krøgli et al., 2018; Schmaltz et al., 2019; Moreno et al., 2025). In addition, modellers could also treat land cover and exposure as dynamic variables to capture long-term changes, such as forest disturbance or construction activities (Pisano et al., 2017; Pittore et al., 2017; Sebald et al., 2019; Pacheco Quevedo et al., 2023). Yet, while many opportunities exist to better represent dynamic processes, early warning applications benefit from parsimony, not least because suitable input data are often limited (Stähli et al., 2015; Kaltenberger et al., 2020; Potter et al., 2021). In our context, relying on scalar







precipitation variables and a date-derived proxy (DOY) strikes a practical balance, ensuring feasibility and computational efficiency, as well as compatibility with the lead times of existing precipitation nowcasts and forecasts.

All models relied on several static spatial variables, with the fall-type model being the most strongly influenced by them (Fig. 5). Consequently, for the fall-type model, it was generally easier to estimate where impacts might occur than when. Given the scale of this investigation, this result is plausible, as potential impact locations for fall-types are generally confined to basins with steep terrain and downslope exposed assets. In contrast, the timing of a fall-type movements is inherently complex to predict, particularly at these scales and when relying solely on meteorological variables. Accordingly, rock fall warning often relies on locally installed sensors or remote sensing techniques that monitor rock faces for precursors of failure (Stähli et al., 2015). For the fall-type model, the two most influential predictors were exposure-related, with the number of buildings clearly dominant. While slope angle might intuitively be expected to dominate at this scale, its effect was largely accounted for by the PPA delineation (Fig. 3).

To move from a purely process-oriented to an impact-based perspective, exposure information was incorporated in two steps, allowing the modelling framework to move beyond conventional spatially explicit landslide early warning approaches (Guzzetti et al., 2020). First, the modelling domain was limited to basins where PPAs coincided with infrastructure. Second, within these basins, exposure variables based on building counts and road networks were included. Across all models, the relationships consistently confirmed that higher exposure increases potential impact. However, a key component of risk, namely the vulnerability of infrastructure to the mass movement processes, is missing in our approach. At this scale, incorporating structural vulnerability is challenging (Caleca et al., 2025), as event data lack damage and hazard intensity information, and the model outputs and exposure data are insufficient to apply damage functions. Furthermore, technical mitigation measures, such as torrent control structures (e.g., check dams and rockfall netting), which are widely implemented in alpine regions to mitigate gravitational natural hazards, were not accounted for in this analysis. By stabilizing slopes and regulating water and sediment flow, these structures play a vital role in reducing hazard intensity and protecting downstream infrastructure. Their absence in the analysis may result in an overestimation of potential impacts (Schlögl et al., 2021).

Besides exposure-related effects, all models showed the highest estimated impact potential in basins with relatively steep terrain (slides and flows) or very steep terrain (falls), particularly in generally drier areas with low mean annual precipitation. The negative relationship with precipitation was most pronounced in the slide- and flow-type models, consistent with previous modelling results (Steger et al., 2024). This suggested that landscapes with higher long-term precipitation (e.g., 2000 mm/year) may be better adapted to wet conditions and therefore less responsive to individual events, unlike drier basins (e.g., 500 mm/year). These results provide quantitative support for the landscape equilibrium hypothesis (Renwick, 1992; Smith et al., 2023).

#### 6.2 Accounting for mass movement runout paths while maintaining a generalized landscape representation

Data preparation aimed to balance landscape generalization with sufficient detail. This involved the delineation of basins and the definition of process-relevant terrain (i.e., PPA) within each. Pixel-based runout information was incorporated into the








basin-based model framework to combine the stability and reduced noise of basin-scale units (Alvioli et al., 2016; Loche et al., 2022; Woodard et al., 2024) with the spatial detail relevant to capture mass movement propagation and impact.

Basins were chosen as the main investigation unit to overcome limitations of high-resolution raster representations. Basin-based units can mitigate uncertainties in landslide positional accuracy, reduce computational demands for large-area prediction tasks, and allow flexible aggregation of input variables (e.g., averages, dominant classes, or coverage proportions) (Alvioli et al., 2016; Woodard et al., 2024). Such units can also reduce the impact of errors or changes in predictor data. For instance, moderately pronounced changes in variables like forest cover or building counts are likely to have little effect on aggregated basin values, whereas in pixel-based models, individual cells and therefore subsequent model predictions may deviate substantially.

However, this increased generalization also has potential drawbacks. Aggregating properties across entire basins can obscure important details, as predictor variables may no longer reflect conditions where processes are most relevant. For instance, phenomena linked to very steep terrain, such as rockfalls, may not be captured by a mean slope over a basin. Similarly, in exposure or risk assessments, assets beyond the actual reach of mass movements may be incorrectly counted as exposed or contributing to risk (Caleca et al., 2025). We argue that pixel-based representations, despite having some disadvantages, also offer advantages in capturing fine-scale spatial variability. This study aimed to combine the benefits of both approaches, preserving sufficient landscape generalization while retaining the spatial detail suitable for capturing mass movement propagation and impact. Delineating PPAs separately for each movement type enabled integration of pixel-based runout information into basin-level representations (Fig. 3), providing a more realistic depiction of process-relevant geoenvironmental conditions and particularly of potentially exposed assets. Nonetheless, the approach remains constrained by uncertainties inherent in the simplified runout modelling procedure and does not replace site- or catchment-specific analyses (Mergili et al., 2015; Wichmann, 2017). In summary, this innovative method mitigates several drawbacks recently highlighted in large-area landslide susceptibility, exposure, and risk assessments (Lima et al., 2021; Loche et al., 2022; Lin et al., 2023; Marchesini et al., 2024; Caleca et al., 2025).

# 6.3 Avoiding oversimplified patterns through tailored sampling

A wealth of studies on data-driven mass movement modelling highlight that the quality of input data is directly reflected in the quality of the resulting models, reinforcing the widely recognized maxim of 'garbage in, garbage out' (van Westen et al., 2008; Petschko et al., 2014; Steger et al., 2016b; Reichenbach et al., 2018; Lima et al., 2022; van Natijne et al., 2023). A less frequently addressed and equally important issue concerns how available input data should be sampled before being included in a data-driven model (Dornik et al., 2022; Rabby et al., 2023; Guo et al., 2024)Data sampling design plays a decisive role in ensuring that models capture relationships beyond those that could be anticipated by common sense. Designing such strategies is a non-trivial task, particularly in the context of space-time modelling, where computational feasibility, data considerations (e.g., group-effects, autocorrelation) and content-related aspects need to be considered (Steger et al., 2024).







In space-time mass movement modelling, sampling design must carefully account for both spatial and temporal dimensions. Spatially, this entails selecting appropriate locations from which training data are drawn. Temporally, sampling must ensure periods are captured that are relevant to the processes under study. If, for example, the present study had relied on a pixel-based approach with random sampling of absences without further restriction, the resulting models would likely have reflected trivial distinctions, separating flat terrain from hillslopes or dry from rainy days, rather than the conditions most relevant for decision-making (Steger et al., 2023, 2024). Previous studies show that such "oversimplified" models may appear to perform well statistically, as distinguishing mass movement presences from randomly drawn, largely irrelevant absences can be straightforward. Guo et al. (2024) showed that sampling absences far from known mass movements, using larger buffer distances, increases dissimilarity between presence and absence observations, thereby boosting measured model performance and suggesting improved model quality. However, from our viewpoint, such findings point to the opposite conclusion: large-distance sampling does not enhance the quality of modelling results, as it systematically forces absences into trivial terrain, artificially inflating performance and promoting oversimplified models with limited decision-making value (Steger and Glade, 2017).

In this study, we avoided sampling in terrain irrelevant to mass movement impacts. Presence and absence observations were restricted to basins where exposed assets (buildings, roads) fall within PPAs, focusing the modelling on areas with realistic impact potential. For instance, fall-type sampling targeted steeper Alpine basins, while slide-type sampling was less constrained, covering much of the Alpine foreland. Similar considerations guided the temporal sampling. To reduce trivial temporal contrasts, sampling was restricted to days with at least moderate rainfall, defined using an arbitrary threshold of 10 mm/day. In our earlier studies (Moreno et al., 2024; Steger et al., 2024), a much lower threshold was used (1.1 mm/day) to include only rainy days while accounting for data uncertainty. Focusing on at least moderate rainfall aimed to better align the models to decision-making contexts, where light precipitation is less relevant. Threshold choice also affects the similarity between presence and absence samples, influencing estimated modelled relationships and model performance scores. For example, limiting absence sampling to heavy rainfall days (>40 mm/day) would likely reduce the discriminative power of short-term precipitation predictors, lowering variable importance and overall model performance. More broadly, the distinction between relevant and irrelevant input data depends on modelling objectives while associated implications apply to both spatial and temporal sampling.

Literature contains numerous examples of exceptionally well-performing data-driven landslide susceptibility models. While there are valid cases of well performing models with appropriate sampling strategies, we assume that in many instances, models rely on oversimplified classification tasks, arising either from unrestricted random sampling or from sampling that even forces absences to overrepresent trivial observations (Zhu et al., 2019; Rabby et al., 2023; Guo et al., 2024). Scepticism toward apparently well-performing landslide models is not new (Hearn and Hart, 2019; Steger et al., 2016a). We assume that, in essence, the root causes of this criticism can often be traced to the underlying sampling design, which leads models to make overly simplistic distinctions that provide limited practical value for decision support. In space-time modelling, this issue can be amplified, as trivial observations may be sampled not only across space, but also over time or simultaneously in both.







# 705 6.4 Model interpretability and model suitability for impact-based warning

Another important aspect for enhancing model utility is interpretability, particularly when results are intended to support decision making. Many machine learning studies in landslide research emphasize performance metrics, often at the expense of geomorphic plausibility and interpretability. This can limit operational uptake, since end-users benefit more from models that are interpretable, plausible, and trustworthy (Lombardo et al., 2020; Collini et al., 2022; Nocentini et al., 2023; Caleca et al., 2024). Using inherently interpretable models or techniques that make results explainable (Maier et al., 2024; Molnar, 2025) is valuable for communicating findings to users and internal plausibility checks, as well as model refinement. From an application perspective, we argue that prioritizing predictive performance is not always ideal, as less performant but more interpretable models can be more useful in practice (cf. Section 6.3). In this study, particular attention was given to model interpretation, using GAMMs to visualize non-linear relationships between predictors and the response, alongside variable importance assessments. Combined with hindcasts and "what-if" scenario analyses, this facilitated a clearer understanding of model response.

In terms of applicability in impact-based warning contexts, the slide- and flow-type models showed clear potential for largearea applications. The fall-type model, by contrast, is less sensitive to short-term variations in weather conditions, thus its
value for precipitation-driven early warning appears limited. Potential operational uptake of the models also depends on the
meteorological input data. All models were trained on a consistent set of inputs derived from CERRA reanalysis (Ridal et al.,
2024). Regarding applicability, it should be noted that precipitation data from reanalyses used to train the models may differ
from the data available in nowcasting or forecasting systems. Such potential biases should be addressed prior to operational
implementation, for example through bias correction or the use of alternative meteorological inputs, to reduce discrepancies
between model training and operational datasets. Ideally, data perfectly consistent with nowcasting or high-resolution
forecasting would be used for model training. In practice, however, such data often lack long-term historical coverage or
exhibit temporal inconsistencies due to changes in operational forecasting systems, limiting their suitability for model training
over extended periods. Nevertheless, building the model on precipitation data with higher-than-daily temporal resolution could
further improve the models, as discussed in Section 6.1.

Another important consideration refers to the interpretation of the spatially transferred model predictions, namely the probability scores. These scores represent conditional probabilities influenced by the class balance in the training data. Consequently, when absences outnumber presences, as in our case, predicted probabilities are shifted toward lower values. Misunderstanding this concept may hinder user uptake and reduce trust, as a value of 0.7 does not imply a 70% chance of a mass movement impact occurring on that day in that basin. In this study, higher scores simply indicate greater similarity to observed impact conditions, while lower scores reflect higher similarity to absence observations. For operational use, these continuous scores would likely need to be converted into warning thresholds (e.g., red, orange, yellow). Thresholding approaches that account for true positive and false alarm rates exist (Steger et al., 2024), but for effective user uptake and


communication, thresholds are best developed in collaboration with end-users and by considering local contexts to ensure tailored decision-support products (Budimir et al., 2025).

The methodological framework presented in this study can be adapted to develop dynamic, spatially explicit impact models

#### 6.5 Transferability of the approach and applications beyond impact-based early warning

resolution spatio-temporal modelling (Benestad et al., 2019; Lazoglou et al., 2024).

for other phenomena and regions. Beyond mass movement impacts, it may also be applicable to other rapid-onset hillslope processes, such as flash floods or snow avalanches, as well as to hazards like wildfires or hail, where impacts result from multiple drivers acting across scales. Key aspects include a tailored yet generalized landscape representation that accounts for spatial uncertainty in input data, for process propagation dynamics, and integration of meteorological, geo-environmental, and 745 exposure/vulnerability information. Targeted sampling to avoid trivial observations is emphasized to ensure meaningful model patterns, while model interpretability can support plausibility checks and effective result communication. Beyond impact-based early warning, this framework can also support the analysis of spatio-temporal patterns and trends over extended areas and periods. For instance, the models developed in this study are intended to produce thousands of daily hindcasts at the basin scale, supporting multi-decadal analyses of patterns. Averaging these hindcasts at the basin level can 750 extend conventional landslide susceptibility and exposure assessments, which typically do not capture the spatio-temporal dynamics of meteorological drivers. A large number of hindcasts can also be used to analyse the frequency of threshold exceedance per basin or to conduct trend analyses, revealing systematic shifts in impact potential over time. In the context of climate impact assessment, the models can, in principle, be adapted to digest climate projection data. Since the models can also be adapted to work with sub-daily inputs, the ongoing tendency in climate modelling toward producing sub-daily 755 information over extended areas and periods (Fosser et al., 2015, 2024) might open avenues for sophisticated climate impact analysis. Nevertheless, potential limitations, such as the "drizzle bias" (overestimation of weak and underestimation of heavy precipitation in climate models) and the spatial skill scale, must be considered when applying climate projection data in high-

## 7 Conclusion

This study introduces a comprehensive, dynamic, and spatially explicit modelling framework for impact-based early warning of different mass movements in the Alpine region. By addressing key limitations of traditional static susceptibility models, the framework integrates geo-environmental, exposure and dynamic meteorological data into a cohesive and robust approach. Models tailored to slide-, flow-, and fall-type movements effectively capture spatio-temporal dynamics and provide interpretable outputs to support informed decision-making. While the slide- and flow-type models show strong potential for operational applications in an early warning context, the fall-type model exhibits limited sensitivity to short-term weather conditions, bearing more the character of a large area, almost static spatial assessment. To the best of our knowledge, these models represent the first globally available impact-based mass movement models, being explicitly process-specific and

https://doi.org/10.5194/egusphere-2025-4940 Preprint. Discussion started: 15 October 2025

© Author(s) 2025. CC BY 4.0 License.

integrating geo-environmental, exposure, and weather information across scales. Their adaptability to other hazards, compatibility with climate data, and applicability across regions highlight the versatility of the underlying methodological framework and its broad utility for disaster risk reduction and climate impact assessments worldwide.

# 8 Supplementary material

Animations for the "what-if" scenarios (Fig. 9) and the hindcasts (Figs. 10 and 11) are available at: https://doi.org/10.6084/m9.figshare.30271795.v1.

#### 775 9 Code and data availability

The R code for model training and basic visualizations, along with the associated training data, the outline of the Alpine Space and basin delineation. will be made publicly available GitHub upon publication: https://github.com/StefanSteger/LandslideImpact AlpineSpace NHESS. To ensure long-term accessibility and proper citation, the GitHub repository will be tagged and assigned a persistent identifier via Zenodo.

#### 780 10 Author contribution

SS: conceptualization, data curation, analysis/modelling, validation, visualisation, writing (original draft). RS: conceptualization, data curation (geo-environmental predictors), validation, writing (review and editing). MM: conceptualization, data curation (basins, exposure), writing (review and editing). SL: curation (CERRA data), writing (review and editing). KE: data curation (impact data), writing (review and editing). AC: conceptualization, data curation CERRA support), writing (review and editing); MS: conceptualization, writing (parts of original draft, review and editing). All authors have read and agreed to the published version of the paper.

#### 11 Acknowledgements

The research leading to these results has received funding from Interreg Alpine Space Program 2021-27 under the project number ASP0100101, "How to adapt to changing weather eXtremes and associated compound and cascading RISKs in the 790 context of Climate Change" (X-RISK-CC).

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
