# Peer review of "Impact-based early warning of mass movements - A dynamic spatial modelling approach for the Alpine region"

_EGUsphere, 2025_

## Author Comment (AC1)

**RC1**: 'Comment on egusphere-2025-4940', Nicola Nocentini, 01 Dec 2025  reply

#

We thank the reviewer Nicola Nocentini (RC1) for the very positive assessment of our work. We are pleased that the manuscript is considered close to being publishable, and we appreciate the recognition of the modelling framework. The manuscript will undergo further revisions in response to comments raised by the second reviewer (RC2), which we believe will further strengthen the contribution.

**Comment regarding the interpretation of the temperature-related predictors:**

We agree that the reviewer raises an important point, and we acknowledge that interpreting modelled relationships, especially for variables that may act as proxies for multiple physical controls, requires particular care.

Indeed, also in our study area, temperature is correlated with elevation, and it is plausible that the temperature predictor may partly capture elevation-dependent spatial patterns. However, we note that in our case the modelling is performed at the basin scale, not at the pixel scale. Individual basins often span a large elevation range, from valley bottoms to crests, implying substantial internal variability. As a result, basin-aggregated temperature values (e.g., one mean temp. value per basin) may often not resolve contrasts between high-alpine areas and lower areas. We point out that lower-elevation areas are much better represented within our study, simply because our analysis explicitly focuses on damage-causing events. High-alpine mass movement processes are therefore not well represented in the impact database, nor captured by our variables. This supports the notion that the measured temperature effect is only partially attributable to elevation-dependent spatial effects. Furthermore, the underlying CERRA data is rather coarse in spatial resolution (5.5 km), which may further limit its ability to capture elevation-dependent temperature effects, as narrow alpine valleys may incorporate a wide range of elevations within one CERRA pixel.

A further and likely more important aspect is that the temperature predictor in our framework is spatio-temporal, meaning that it varies not only across space but also across time (i.e., each basin receives a value for each sampling day). Consequently, temporal variability in temperature may contribute substantially to the model response. Seasonal variations are already explicitly captured through the day-of-year (DOY) predictor, highlighting that temperature is intended to account for a different effect, as noted in **LINE 459:** *"Seasonal variation, modelled using a circular DOY effect, indicated that periods with reduced vegetation effects (i.e., winter, early spring) were associated with higher impact probabilities (Fig. 6g), noting that temperature and precipitation effects are accounted for by other variables in the model."*

In summary, this led us to interpret the "remaining" temperature effect as likely reflecting short-term meteorological influences. For example, on a summer day (DOY ~180) with heavy rainfall, convective activity is more likely than on a winter day with similar daily rainfall amount. This interpretation aligns with the literature indicating that high temperatures are associated with an increased potential for convective activity (see cited literature Giorgi et al., 2016). Given this, interpretation should still be provided with caution, acknowledging that confounding between predictors (e.g., temperature, elevation, and DOY) may influence modelled relationships. In the revised manuscript, we will therefore be more careful in our formulations to explicitly address this point and refine the interpretation. For instance:

**LINE 588:** *"In this context, daily precipitation and temperature variables represented …"* will be revised to: *"In this context, daily precipitation and temperature variables were assumed to represent …"*

In the Methods section, we will now include further information on our intention to use temperature by revising **LINE 317** from: *"Daily mean temperatures and two binary indicators were used to represent*

*potential temperature effects."* to *"Daily mean temperatures and two binary indicators were used to mainly represent potential temporal temperature effects, while we emphasize that spatial temperature effects (i.e., differences between high elevations and valley bottoms) were likely to be only partially captured due to the basin-based landscape representation and the coarse resolution of the underlying CERRA data."*

Additional text in the Discussion section will now further clarify why data-driven model interpretation is challenging, highlighting potential confounding between predictors. We will add to **LINE 591**: *"However, it should be noted that the interpretation of individual predictor effects remains challenging, as confounding between variables (e.g., temperature, elevation, and DOY), along with specific biases in landslide inventories, can strongly influence the measured effects and limit a straightforward physical attribution, as discussed in Steger et al. (2021)."*

Steger, S., Mair, V., Kofler, C., Pittore, M., Zebisch, M., and Schneiderbauer, S.: Correlation does not imply geomorphic causation in data-driven landslide susceptibility modelling – Benefits of exploring landslide data collection effects, Science of The Total Environment, 776, 145935, https://doi.org/10.1016/j.scitotenv.2021.145935, 2021

Finally, we will now emphasize in the Discussion that incorporating specific weather types may reduce the models' reliance on temperature by adding the following to LINE 597:

*"Incorporating information on specific precipitation and weather types to explicitly distinguish convective events from other weather dynamics could further improve the representation of rainfall-triggering conditions, making the models less reliant on proxies, such as dynamic temperature effects."*

---

## Author Comment (AC2)

**RC2**: 'Comment on egusphere-2025-4940', Anonymous Referee #2, 10 Dec 2025  reply

- **Reviewer original comment in blue**
- **Authors answer in black**

**1 Dear Authors,**

I have read and carefully evaluated the manuscript "Impact-based early warning of mass movements – A dynamic spatial modelling approach for the Alpine region" and I believe it deserves publication after minor revisions. The topic is well within the aims and scopes of NHESS journal, the research is original, significant and well written.

We thank the reviewer (RC2) for the careful review of our manuscript, as well as for the positive assessment and the numerous constructive suggestions for improvement. We are pleased that the reviewer considers the manuscript suitable for publication after minor revisions.

**2 GENERAL COMMENTS**

I think that the manuscript is affected by a relevant limitation that has not been clearly stated in the text. I suggest to openly present and discuss it: the work is very good, and this limitation does not diminish its potential, but it makes better framing necessary. This limitation concerns the input landslide dataset. If I have understood correctly (a figure would have made it clearer – see one of my specific comments), the landslide used are only in Austria and in South-Tyrol, which is an Italian Province if I'm not mistaken. This is a big limitation, because you try to apply your model on an area that is very bigger than the area where it is trained.

We thank the reviewer for highlighting this important point. We agree that the spatial extent of the input landslide dataset represents a relevant limitation that should be discussed more explicitly. As correctly noted by the reviewer, the inventory used for model training is restricted to Austria and South Tyrol (see revised Figure 1), while the model is applied to a larger area. This raises concerns regarding spatial transferability.

In the originally submitted manuscript, we attempted to address this limitation through careful model design and variable selection, with the aim of facilitating the transfer of modelled relationships beyond the training area. Several methodological choices were made to promote generalization. These include limiting the maximum allowed flexibility of smooth terms, adopting generalized spatial units (i.e. basin-based, rather than pixel-based), and reclassifying input data to ensure consistent representation of predictor variables across the entire area.

For example, the number of land cover and lithology classes was substantially reduced through reclassification (see **LINE 298f**). This reclassification was done to ensure that each class occurring within the Alpine Space is well represented in the training area, while avoiding classes that occur only outside the training region. As a result, the selected land cover and lithological classes are both well represented in the training dataset and broadly distributed across the Alpine Space (see map in answer to #13 below).

Under these circumstances, and as formerly stated in **LINE 187f**, we considered the training area to be *"representative of the environmental variability across the Alpine Space."* The training region encompasses a wide range of environmental conditions, including highly diverse landscape characteristics and elevation gradients from approximately 200 to nearly 4000 m a.s.l., while the underlying predictor variables are consistently generalized across the study domain. Together with the

above-mentioned generalization steps, we believe this provides a reasonable, though not exhaustive, basis for broader application of the model. Nevertheless, we acknowledge that limitations remain, and we agree that these should be made more explicit. To address this, in the revised manuscript we will:

- Explicitly visualize the training area also in the revised Figure 1 (plus adding a new Figure 4; see answers to #12 and #15).
- Modify the title from *"Impact-based early warning of mass movements – A dynamic spatial modelling approach for the Alpine region"* to *"Dynamic spatial modelling of mass movement impacts for large areas: A data-driven framework for impact-based early warning"*, to account for several other raised issues by the reviewer (see also answer to #7).
- More explicitly discuss the limitations associated with extrapolating modelled relationships beyond the training area and the lack of quantitative validation outside this area, while also reworking the narrative to better highlight the limitations (for details see: #3, #4, #5, #6, #7)

We believe these revisions will provide a more balanced framing of the study while preserving the validity and relevance of the presented results.

**3 Specific problems are:**

- Technically, you are calibrating the model in a "core area" and trying to apply it to a wider surrounding area. Technically, this is more an extrapolation than an application.

We agree that the term *"application"* may be interpreted ambiguously in this context. To avoid potential misunderstandings, we will revise the manuscript to consistently use a more precise terminology, such as *"spatial transfer"* or *"spatial extrapolation of modelled relationships."* For example, **LINE 385**, amongst others, will be revised from: *"A series of predictive maps and animations were generated by applying the three models across the Alpine Space"* to: *"A series of predictive maps and animations were generated by spatially extrapolating the modelled relationships of the three models across the Alpine Space."*

**4 You cannot measure the reliability of this extrapolation, because you don't have landslide data to validate your extrapolation.**

We agree. Beyond the qualitative evaluation performed using selected hindcast events (e.g., the Vaia storm), we are currently unable to quantitatively assess the performance of the spatial extrapolation due to the lack of suitable landslide impact data outside the training area. As such, the quality of the extrapolated results cannot be measured quantitatively. We will explicitly acknowledge this limitation in the revised manuscript. Specifically, we will state this constraint in the Methods section (Section 4.4.1) and further discuss its implications in the revised Discussion section (Section 6.4).

Adding to **LINE 370**: *"It should be noted that the model performance assessment was restricted to the training region (Fig. 1, 4) due to the lack of independent landslide impact data."*

Adding to **LINE 728**: *"An additional limitation for operational application is that the performance of the spatially transferred model predictions could not be quantitatively assessed beyond the model domain (Fig. 1) due to the lack of time-stamped landslide data containing explicit information on infrastructure impacts. Qualitative hindcast analyses, however, provide indication of model plausibility, while future refinements and quantitative evaluation could be achieved using spatially consistent impact data."*

**5 The claim that the core area (located in the central Apls) is representative of all the Alpine area is not convincing: only taking into account geology, the characteristics are very different with the Western part (harder metamorphic rocks) or the Eastern part (dominated by karst landscapes).**

As noted also in #2, several steps were taken to promote generalization beyond the training area. Austria and South Tyrol cover a wide range of landscapes and geological settings, from low-elevation lowlands in the far east of Austria, to undulating hilly landscapes in the areas surrounding Lower and Upper Austria, to typical high Alpine regions with steep valleys in the west, including a variety of lithologies (see map in #13). Generalization of land cover and lithology classes ensured that only features represented within the training area were also present outside it. This generalization, however, does not allow the model to capture very specific local characteristics, such as unique lithologies occurring only outside the training area. We acknowledge that limitations related to regional specificities remain and will be discussed in the revised version. We will soften the statement in **LINE 186** and will clearly highlight the associated limitations:

*"The datasets cover the entire Alpine Space, while mass movement event data for model training was available for Austria and South Tyrol, a region considered reasonably representative of the environmental variability across the Alpine Space. This training area comprises 3,696 basins (Fig. 1, 4) and, in addition to time-stamped impact data, offers substantial diversity in geology, topography, land cover and climatic features, providing a robust basis for developing a model with spatial transferability across the Alpine Space. The generalization of input variables, including the reclassification of lithology and land cover (cf. Section 4.2.3), nonetheless has inherent limitations and does not allow the model to capture specific local characteristics, such as unique lithologies present only outside the training area."*

**6 I don't know why you limited your dataset only to the core area. Open landslide data (with timestamp) are available for Italy**

Peruccacci, S., Gariano, S. L., Melillo, M., Solimano, M., Guzzetti, F., & Brunetti, M. T. (2023). ITALICA, an extensive and accurate spatio-temporal catalogue of rainfall-induced landslides in Italy. *Earth System Science Data Discussions*, *2023*, 1-24.

Calvello, M., & Pecoraro, G. (2018). FraneItalia: a catalog of recent Italian landslides. *Geoenvironmental Disasters*, *5*(1), 13.

And also in Slovenia and Switzerland nation-wide inventories are available (although not open-access).

We thank the reviewer for providing these references, which will be included in the revised manuscript when discussing the availability and limitations of landslide inventory data. We note, however, that our study focuses explicitly on an impact-based perspective, specifically on time-stamped landslide events with documented impacts on infrastructure (data that systematically covers more than a decade rather than selected or event-based compilations). For this reason, as well as due to resource limitations, the broader datasets mentioned by the reviewer were not included, since our analysis required detailed information on process type, daily timing, spatial accuracy, and explicit documentation of infrastructure damage. Although we did not include landslide inventories from across the entire Alpine Space in this study, we agree with the reviewer that future efforts should be made to harmonise the data (e.g. by using consistent categories for landslide type, damage to infrastructure and collection method). This would be an important step towards improved model performance, allowing specific local geomorphic and climatic characteristics to be accounted for.

In more detail, the IFFI data used in our study pertains specifically to the South Tyrolean IFFI version (cf. Fig. 1). IFFI inventories can vary substantially across regions. Our previous collaborations with local IFFI data managers (e.g., Steger et al., 2021) enabled access to additional input data beyond what is officially available via IdroGeo (https://beta.idrogeo.isprambiente.it/app/), including explicit information on infrastructure impacts, information not (openly) available for other IFFI datasets. A similar situation applies to the Austrian datasets, where detailed knowledge of data quality and attributes (e.g., Georios, managed by GeoSphere; WLK, through previous project collaborations) allowed us to filter/select data suitable for an impact-based modelling approach. To avoid repeating previously reported findings, we will refer the reader to our earlier publications rather than provide an extensive discussion of inventoryrelated issues that were already addressed before (e.g., see cited references in **LINE 664f**). Nonetheless, in the revised version, "inventory and associated data sampling issues" will be discussed more thoroughly in the context of our study within the revised Section 6 (see also the answer to #26 for details on why deeper insight into inventory properties and attributes was considered important for our purpose).

**7 Those limitations should be clearly highlighted in the manuscript. Moreover, I think that the narrative of the article should be reworked. From the title and abstract, I was expecting a warning system running on the whole Alpine region. You should be clearer on that. The framework you propose is for the Alpine region, but due to limited data you calibrate it only in a smaller portion of the area. You try to present an application to the whole region, but it's more like an exemplification of the potentiality of the approach, you actually don't have data to validate it (the comparison with Vaia storm cannot be considered a proper validation – see the specific comment).**

As mentioned above, we will modify the manuscript to better highlight the limitations of our work and to handle expectations regarding its scope. Furthermore, we will update the manuscript title to *"Dynamic spatial modelling of mass movement impacts for large areas: A data-driven framework for impact-based early warning"*. This change better reflects the modelling and framework focus and clarifies that the study is not a fully operational Alpine-wide system. We will also revise the abstract to emphasize that the core contribution lies in the flexible modelling approach itself, rather than in delivering a fully operational warning system. In the revised abstract, we will explicitly mention that:

*"This study introduces a flexible, dynamic, and spatially explicit modelling framework for supporting impact-based early warning of precipitation-induced mass movement processes. The framework is (…)"*

Later in the abstract, we will then also clarify that:

*"The primary contribution of this research lies in demonstrating a transferable and generalizable modelling framework, rather than delivering a fully operational warning system."*

**8 (2) I think the existence of operational or prototypal landslide warning systems in your study area is not properly accounted for. In Italy, Switzerland, Slovenia, some examples do exist. Some of them also try to better account for the possible impacts (and not just a mere probability of occurrence). I think it would be fair to account for similar attempts in your study area before you move to describe the improvement you propose.**

Wicki, A., Lehmann, P., Hauck, C., Seneviratne, S. I., Waldner, P., & Stähli, M. (2020). Assessing the potential of soil moisture measurements for regional landslide early warning. *Landslides*, *17*(8), 1881-1896.

Segoni, S., Nocentini, N., Barbadori, F., Medici, C., Gatto, A., Rosi, A., & Casagli, N. (2025). A novel prototype national-scale landslide nowcasting system for Italy combining rainfall thresholds and risk indicators. *Landslides*, *22*(5), 1341-1366.

Tiranti, D., & Ronchi, C. (2023). Climate change impacts on shallow landslide events and on the performance of the regional shallow landslide early warning system of Piemonte (Northwestern Italy). *GeoHazards*, *4*(4).

Auflič, M. J., Šinigoj, J., Krivic, M., Podboj, M., Peternel, T., & Komac, M. (2016). Landslide prediction system for rainfall induced landslides in Slovenia (Masprem). *Geologija*, *59*(2), 259-271.

Similarly, outside the study area some relevant examples do exist, like:

Krøgli, I. K., Devoli, G., Colleuille, H., Boje, S., Sund, M., & Engen, I. K. (2018). The Norwegian forecasting and warning service for rainfall-and snowmelt-induced landslides. *Natural hazards and earth system sciences*, *18*(5), 1427-1450.

We thank the reviewer for highlighting these relevant systems. We will account for these examples in the revised manuscript to provide proper context and to acknowledge previous efforts. We note, however, that while some of these systems incorporate aspects of potential impacts, impact data is not central to their design, and therefore they are generally not impact-based in the strict sense as defined in our study. In the introduction, we will now include:

*"Several examples of operational or prototypal landslide warning systems exist in the study area, including examples from Italy, Switzerland, and Slovenia (Auflič et al., 2016; Wicki et al., 2020; Tiranti and Ronchi, 2023; Segoni et al., 2025), as well as in other countries such as Norway (Krøgli et al., 2018). However, even where operational systems exist, they are generally considered complex and uncertain, while their underlying modelling framework mainly focuses on process prediction, not impact (Alfieri et al., 2012; Piciullo et al., 2018; Guzzetti et al., 2020)."*

SPECIFIC COMMENTS

**9 L22. Please cut "Beyond accounting for meteorological, geo-environmental, and exposure nformation,". It is a repetition and the text flows nicely even without it.**

 This will be done.

**10 L39-45. Impact-oriented, impact-based, meteorologically focused… the use of all these terms is confusing. Can you clearly define them providing references?**

From our experience, the terms "impact-oriented" and "impact-based" warnings are often used interchangeably, particularly in practice within national meteorological services. This observation comes also from our work as member of the MeteoAlarm (https://www.meteoalarm.org/en/live/) IbW Expert Team. However, there are important distinctions, which we highlight in the introduction, to differentiate impact-oriented warnings (e.g., current practice at GeoSphere Austria) from impact-based warnings (envisaged practice). To improve clarity, we will simplify this section, add supporting references, and avoid the term "meteorologically-focused" to make the discussion easier to follow (**LINE 36f**).

*"While effective in identifying potentially hazardous weather conditions in advance, traditional warnings do not explicitly address the negative consequences that may follow. To improve risk management and promote more appropriate responses, it is essential to assess how critical weather conditions may impact lives, livelihoods, and property (WMO, 2015; Casteel, 2016; Weyrich et al., 2020). Accordingly, several national meteorological and hydrological services now aim to account for the potential impacts of weather phenomena to better inform decision-makers and the public (Uccellini and Hoeve, 2019; Kaltenberger et al., 2020; Potter et al., 2021; Sivle et al., 2022).*

*Impact-oriented approaches primarily rely on meteorological information, emphasizing expected weather conditions while using expert judgment to infer potential impacts. In contrast, impact-based approaches offer a more quantitative assessment of potential consequences by explicitly integrating meteorological data with geo-environmental and socio-economic data, including exposure and vulnerability (Kaltenberger et al., 2020; Potter et al., 2021). The transition toward impact-based warnings is encouraged by the World Meteorological Organization and international frameworks, such as the Sendai Framework for Disaster Risk Reduction and the United Nations Early Warnings for All initiative, which advocate for integrated, risk-informed strategies (WMO, 2015)."*

**11 L60 – "short term disturbances": isn't it cleared to write "external triggers", as opposed to the abovementioned static predisposing factors and preparatory factors.**

We agree. It will be changed to "external triggers".

**12 Fig 1: I think the elevation in the bar should go from zero to 4810**

Yes, the highest elevation in the Alpine Space exceeds 4,800 m (Mont Blanc). However, in Fig. 1 the elevation colour scale was intentionally capped, with elevations above 3,000 m mapped to white. From our viewpoint, this design choice enhances visual differentiation between hilly and mountainous regions and areas of high-alpine terrain. If the colour scale were stretched to the maximum elevation (≈4,810 m), the color gradient would be compressed, and most high-alpine areas around 3000 m would appear similar to mid-elevation terrain (e.g., 1500) in brownish colour, resulting in little visually discernible distinction. The purpose of this colour selection was to highlight the presence and spatial distribution of high-alpine terrain in the western part of the training area (see revised Fig. 1 below). Using the full elevation range would obscure these features rather than clarify them. The revised Figure 1 will now use an elevation range from 0 to > 3,000 m and will additionally include the training area boundaries, shown in red and blue:

[Figure]

**13 Study area: I think a geological or lithological map is very important to be shown.**

We agree that lithology is an important controlling factor for mass movement processes. However, we prefer not to include additional lithological maps in the manuscript for several reasons. First, the manuscript is already relatively voluminous, and a meaningful representation would require three separate lithology maps, one for each process-type, to accurately reflect the input data used in the modelling framework. For reference, an illustration of the slide-type lithology map (dominant class within slide-type PPAs), derived directly from the open and shared basin dataset, is shown below.

[Figure]

Furthermore, the lithological data used in this study are not original products of this work, but are derived from published sources and can be consulted in the cited references (Donnini et al., 2020):

Donnini, M., Marchesini, I., and Zucchini, A.: A new Alpine geo-lithological map (Alpine-Geo-LiM) and global carbon cycle implications, GSA Bulletin, 132, 2004–2022, https://doi.org/10.1130/B35236.1, 2020.

In addition, the variable importance assessment indicates that, for the models most relevant for our study context (e.g., slide-type and flow-type), lithology plays a relatively minor role compared to other predictors (cf. Fig. 5). Including lithological maps would therefore create consistency issues, as similar visualizations would then be expected for other, more influential variables.

Additionally, regarding the interpretation of "lithological effects," we refer to the revised Discussion section 6.1, which addresses the potential for confounding. For example, topography may partially capture lithological effects due to their correlation:

*"However, it should be noted that the interpretation of individual predictor effects remains challenging, as confounding between variables (…), can strongly influence the measured effects and limit a straightforward physical attribution (…)"*

To ensure transparency, all input data, including basin delineations and the full set of 42 static variables are available via the cited repository, enabling interested readers to explore the (lithological) information in detail. To guide the reader, we will add the following sentence at the end of the study area section:

*"A visual overview of geo-environmental variables (e.g., topography, land cover, lithology), mean annual precipitation, and exposure-related features (e.g., buildings, roads) across the study area can be explored through the basin-based dataset provided in the repository."*

**14 L161: please consider adding also cultural heritage.**

Will be done (L161): *"Across the Alps, various elements at risk are exposed, including population, infrastructure, cultural heritage, and critical assets."*

**15 DATA section: It is quite important that you show in a map the landslide data used by the model. This is standard practice in landslide studies, moreover it is very important in your case of study because it should be clear that the dataset used covers only a very specific section of the study area.**

We agree and will add the following figure and caption to the manuscript.

[Figure]

**Figure 1 Map of the training area showing presence locations for each movement type. Basins with slide-type presence observations comprise 1,014 spatial units, flow-type 691 basins, and fall-type 549 basins. Note that the number of presence basins is lower than the total number of presence observations used for modelling, since multiple days with registered mass movement occurrences may have been recorded within a basin during the 16-year period.**

**16 L187: I strongly disagree with this claim of representativeness. Landforms and lithology are very different in other parts of the Alps (see also my first general comment). I strongly recommend that you clearly state this as a main limitation of the work instead of minimizing it. The approach you present is very good and the paper deserves publication anyway. No need to pretend that everything is perfect. Perfection does not exist in research, otherwise what's the point in progressing further?**

Thank you, we agree. As detailed in #5, we will soften the statement and explicitly highlight the associated limitations in the revised manuscript. Several new text passages will be added in response to reviewer comments to clarify this issue and address concerns regarding representativeness.

**17 L196: writing "from Northern Italy" is confusing: you actually use data from a single province among the many provinces/regions of Northern Italy. Please, be clearer.**

Will be done by adding:

**LINE 186**: *"The datasets cover the entire Alpine Space, while mass movement event data for model training were available for Austria and the Italian province South Tyrol (Fig. 4)"*

**LINE 196**: *"The landslide data for South Tyrol, including slides, flows, and falls, originate from the provincial implementation of the national landslide inventory IFFI (Inventario dei Fenomeni Franosi in Italia), which contains explicit attributes on recorded impacts. IFFI data (…)"*

**18 Figure 3: I suggest adding legend to all panels: having self-explanatory figures is always better than relying on long captions.**

We agree. The revised figure will look as follows:

[Figure]

**20 L247-252. I suggest moving this part some lines above. I wrote a comment at line L228 about the definition of the starting points, then I discovered that they were defined here.**

We agree and will move this section to an earlier position in the manuscript so that the definition of the release areas is introduced before the process path simulations.

**21 L266. I see your point and I agree, but can't you find some supporting basis to give a stronger background to this threshold?**

We agree that a stronger background would be desirable. We therefore deliberately refer to the selected value as an *"arbitrary threshold"* to emphasize that it was chosen pragmatically to represent a moderate precipitation day, with the aim of excluding light rainfall amounts that are unlikely to be relevant in a warning or decision-making context. At the same time, there is no widely accepted classification scheme for daily accumulated precipitation, as most intensity classifications focus on hourly rates and remain inhomogeneous, lacking standardized nomenclature (cf. Dunkerley, 2021).

Dunkerley, D. L.: Light and low-intensity rainfalls: A review of their classification, occurrence, and importance in landsurface, ecological and environmental processes, Earth-Science Reviews, 214, 103529, https://doi.org/10.1016/j.earscirev.2021.103529, 2021.

Being aware of this issue, we explicitly address and critically reflect on this choice later in the Discussion (**LINE 690f**), where its implications for model behaviour and performance are discussed in detail:

*"Focusing on at least moderate rainfall aimed to better align the models to decision-making contexts, where light precipitation is less relevant. Threshold choice also affects the similarity between presence and absence samples, influencing estimated modelled relationships and model performance scores. For example, limiting absence sampling to heavy rainfall days (>40 mm/day) would likely reduce the discriminative power of short-term precipitation predictors, lowering variable importance and overall model performance. More broadly, the distinction between relevant and irrelevant input data depends on modelling objectives while associated implications apply to both spatial and temporal sampling."*

**22 L277: again, I like the approach but just to be sure: aren't you afraid to omit some potentially relevant interactions between subsequent storms?**

We would like to clarify that the 30-day gap applies only to the sampling of absence days within the same basin, where it was introduced to reduce temporal dependence between absence samples. This constraint does not apply to presence observations. Consequently, multi-day storm events are fully retained in the dataset if they triggered multiple landslides in a region over consecutive days. The purpose of the absence sampling was to represent independent moderate rainfall days without reported impacts. As our modelling approach is data-driven and aims to generalize modelled relationships (see previous answers), we do not attempt to capture specific interaction effects or process cascades. We believe, capturing such detailed event-specific dynamics would likely require a more tailored, less generalized approach that may raise further concerns regarding the models spatial and temporal transferability.

**23 L284. I think it is normal that you have a different number of presence observations for each landslide type. Beside that, a downside of your approach that you could have easily controlled is that for each landslide type you have a different presence/absence ratio. Therefore, your model is not uniformly balanced with respect to the different landslide type. Maybe it is worth mentioning this in the discussion of the main limitations.**

We agree that sampling is an important topic. We would like to clarify that sampling in a spatio-temporal context is more complex than purely spatial sampling, as commonly done in landslide susceptibility studies. In our study, we initially performed sampling in space with a 1:1 ratio at the basin level, as described in **LINE 273**: *"This was done while maintaining a one-to-one ratio between presence and absence basins for each movement type (…)"*. Thus, in space, presence and absence locations are equally represented.

Sampling in time, however, is then more challenging. To reduce temporal autocorrelation, we excluded absence dates within a 30-day gap in the same basin, while also aiming to capture season-related effects (e.g., day-of-year, temperature etc.). For this purpose, the initial absence sampling dates were, as described in **LINE 279**, distributed *"…evenly across months and years, ensuring equal representation of all time periods, in analogy to Steger et al. (2023)."*

In simple terms, this ensured that each day of the year was equally represented by the absence sample. However, because we focused on moderate rainfall days, we then excluded from this evenly distributed temporal sample any observations below the 10 mm precipitation threshold (as described in **LINE 280f**).

The figure below illustrates this process. On the left, counts of absences are shown after excluding absence dates with a gap smaller than 30 days within a basin and after downsampling days per month

to ensure equal monthly representation (but before applying the precipitation filter). On the right, the effect of the rainfall filter is shown: more absences remain in summer, reflecting the higher frequency of rainy days. This provided a representative distribution of rainy days across the year in our absence sample, allowing the model to capture effects such as seasonality.

[Figure]

If we had enforced an equal number of presences and absences across time, seasonal effects or temperature effects could not be captured, because winter and summer days, as well as cold and warm temperatures, would be represented equally in both presence and absence samples, leaving no discriminative power for these variables.

As this paper builds upon our earlier work where these sampling issues are described in more detail, we refer the reader to our previous publications for further details (**LINE 280**: "in analogy to Steger et al. (2023)"). Nonetheless, since we consider sampling crucial, we dedicate an entire discussion section to this topic. See revised version of *"6.3 Avoiding oversimplified patterns through tailored sampling"*.

Finally, the reviewer is correct that this sampling strategy results in different presence/absence ratios between the models, which is why the resulting probability scores cannot be directly compared across models. Consequently, we have added the associated limitations to the Discussion section. The revised version more explicitly highlights the underlying sampling issue and will now read as follows (**LINE 729f**):

*"Another important consideration refers to the interpretation of the spatially transferred model predictions, namely the probability scores. These scores represent conditional probabilities influenced by the class balance in the training data. As a result, when absence observations outnumber presences, as is our case, predicted probabilities are generally shifted toward lower values. This also means that the raw probability scores from the three models are not directly comparable, as they are based on different presence-to-absence ratios. Misunderstanding this concept may hinder user uptake and reduce trust, as a value of 0.7 does not imply a 70% chance of a mass movement impact occurring on that day in that basin. In this study, higher scores simply indicate greater similarity to observed impact conditions, while lower scores reflect higher similarity to absence observations. For operational use, these continuous scores would likely need to be converted into warning thresholds (e.g., red, orange, yellow). Thresholding approaches (…)"*

#24 Table 1. It would be nice to have a figure at least for the static variables. It could be placed in an annex.

We would prefer not to create dozens of plots for the many static variables (i.e., the same variable differs across the three movement types due to variation in the associated PPA). Instead, we would favour to forward readers to the repository, where the full set of 42 static variables can be inspected. We will add to the Study area section: *"A visual overview of geo-environmental variables (e.g., topography, land cover, lithology), mean annual precipitation, and exposure-related features (e.g., buildings, roads) across the study area can be explored through the basin-based dataset provided in the repository."*

#25 L408 – Interestingly, rockfall is the least-performing category and it is also the one with the least numerous sample and the most unbalanced presence/absence sample

It is true that the fall-type model shows the lowest model performance and is also characterized by the smallest sample size and the most unbalanced presence/absence ratio. These aspects may contribute to reduced performance. Due to the still relatively large sample size (n=4,069; 647 presences) and the generalized modelled relationships, however, we still consider the fall-type model behaviour to be primarily driven by the process-related factors discussed in the manuscript, for example in **LINE 609f**:

*"Consequently, for the fall-type model, it was generally easier to estimate where impacts might occur than when. Given the scale of this investigation, this result is plausible, as potential impact locations for fall-types are generally confined to basins with steep terrain and downslope exposed assets. In contrast, the timing of a fall-type movements is inherently complex to predict, particularly at these scales and when relying solely on meteorological variables."*

In the revised Discussion, we will now more explicitly acknowledge alternative influences on the lower model performance by adding the following statement to **LINE 718f**:

*"The fall-type model, in contrast, is less sensitive to short-term variations in weather conditions. Consequently, its utility for precipitation-driven early warning appears limited. The comparatively small number of presence observations, along with the more pronounced class imbalance between presences and absences, may further contribute to the reduced performance of the fall-type model."*

#26 Fig 5. I was wondering that the importance of the urban environment maybe reflects the fact that it is responsible both for the exposure to risk and the increase of hazard (e.g. buildings and infrastructures in a mountain environment could alter the stability of the slopes). In a few words, buildings and infrastructures contribute to an increase in risk twice: they increase the hazard and the increase the exposure. What do you think about it? Could it be reasonable?

This is an important and interesting issue that we already addressed in detail in Steger et al. (2021). Several of these aspects are indeed true. However, data-driven methods may not be able to distinguish them, as *"more landslides near infrastructure"* can mainly result from three factors: (i) underreporting of landslides far from infrastructure, (ii) increased occurrence probability due to human interventions and construction activities near infrastructure, and (iii) higher damage potential due to valuable assets, which may in turn lead to more systematic reporting (point i). Based on our experience with the inventories used, we are confident that the raw unfiltered landslide data is strongly affected by a reporting bias and relatively systematic when it comes to infrastructure damage.

In fact, we explicitly excluded movements (**LINE 260**) *"lacking documented infrastructure impact"* and the delineation of PPAs further focused our analysis on areas potentially prone to infrastructure damage. Our approach follows Steger et al. (2021), where landslide data representing damage events were used

to identify areas prone to damaging or infrastructure-threatening landslides. By shifting the focus from general susceptibility ("process perspective") to damaging landslides ("impact perspective"), inherent data biases were no longer detrimental but became informative, helping the model capture patterns relevant to infrastructure impact. Similarly, in our study, building and road variables were assumed to primarily represent "exposure effects." Given the already voluminous nature of the paper, we will refer readers to our earlier publication for a deeper discussion of these issues. In the revised manuscript (Section 6.1), we will add:

*"(…) First, the modelling domain was limited to basins where PPAs coincided with infrastructure. Second, within these basins, exposure variables based on building counts and road networks were included. Across all models, the relationships consistently showed that higher exposure increases potential impact. However, despite careful efforts to prepare representative impact data, we cannot fully exclude the possibility that the estimated exposure effects are partly confounded by inventory-related biases or by elevated landslide occurrence in the vicinity of infrastructure resulting from increased human interventions and construction activities (Steger et al., 2021)"*

Steger, S., Mair, V., Kofler, C., Pittore, M., Zebisch, M., and Schneiderbauer, S.: Correlation does not imply geomorphic causation in data-driven landslide susceptibility modelling – Benefits of exploring landslide data collection effects, Science of The Total Environment, 776, 145935, https://doi.org/10.1016/j.scitotenv.2021.145935, 2021.

**27 L520 In my opinion the potential has not been demonstrated. You showcased a possible application, but to demonstrate the application you need a validation and you don't have the data. We do not know how many false alarms and missed alarms you would have issued with the maps shown in fig. 10. The comparison you present with damages aggregated at the district level are of little use compared to the objective and the granularity of the proposed warning system. I think it is fair to say that you show that the proposed framework could be used to produce output maps but work (including data collection) is needed to validate the approach and fully demonstrate the applicability to real-case scenarios.**

As also noted in several of our responses above, we agree with this assessment and will revise the section to more clearly highlight the limitations of our work. In this specific case, we will rephrase the sentence as follows: *"Hindcasting was used to analyse model predictions under known conditions and to illustrate the output of the proposed framework."*

**28 Section 6.4 Maybe there is another point to discuss: thinking about real-time applications, the integration of real-time rainfall data, the dataflow, and the processing time are fundamental issues.**

Thank you for this suggestion. This is exactly what we aim to address in our recently started project "PRE4IMPACT-AT" (see Factsheet at https://projekte.ffg.at/projekt/5136099), which will focus specifically on Austria. We will add the following text to the Discussion in Section 6.4:

*"Furthermore, real-time application would additionally require an automated workflow to integrate near-real-time meteorological data and deliver timely model outputs. While this was beyond the scope of the present study, it represents an important step for future work and operational uptake. An additional limitation for operational application is that the performance of the spatially transferred model predictions could not be quantitatively assessed beyond the spatial training domain (Fig. 1) due to the lack of time-stamped landslide data containing explicit information on infrastructure impacts. Qualitative hindcast analyses, however, provide some indication of model plausibility, while future refinements and quantitative evaluation could be achieved using spatially consistent impact data across the entire area."*

**29 Discussion – I suggest synthetizing it. Everything is correct and interesting, but many things were already stated in advance, so this part has a lot of repetitions and the attention of the reader may decline.**

While we have added new content in response to reviewer requests, we will rework the Discussion section to synthesize the material and shorten it where possible, in order to reduce repetitions and maintain reader engagement.

#30 Discussion – I suggest adding a subsection about the main limitation of the work (or maybe 6.5 may be the right place to comment on the main limitations I identified).

In response to the reviewer suggestions, we will add new text passages on the main limitations of our work, placing them where they fit best in the Discussion. Most of the newly added limitations will be included in Section "6.4, *Model interpretability and model suitability for impact-based warning*", as they primarily concern the practical suitability for operational uptake and the limitations of a missing quantitative model validation beyond the training area.